# A Semi-Lagrangian Godunov-Type Method without Numerical Viscosity for Shocks

Valeriy Nikonov 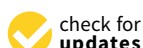

Aircraft Construction and Design Department, Samara University, Moskovskoye Shosse, 34, 443086 Samara, Russia; nikonov.vv@ssau.ru

**Abstract:** One of the most important and complex effects in compressible fluid flow simulation is a shock-capturing mechanism. Numerous high-resolution Euler-type methods have been proposed to resolve smooth flow scales accurately and to capture the discontinuities simultaneously. One of the disadvantages of these methods is a numerical viscosity for shocks. In the shock, the flow parameters change abruptly at a distance equal to the mean free path of a gas molecule, which is much smaller than the cell size of the computational grid. Due to the numerical viscosity, the aforementioned Euler-type methods stretch the parameter change in the shock over few grid cells. We introduce a semi-Lagrangian Godunov-type method without numerical viscosity for shocks. Another well-known approach is a method of characteristics that has no numerical viscosity and uses the Riemann invariants or solvers for water hammer phenomenon modeling, but in its formulation the convective terms are typically neglected. We use a similar approach to solve the one-dimensional adiabatic gas dynamics equations, but we split the equations into parts describing convection and acoustic processes separately, with corresponding different time steps. When we are looking for the solution to the one-dimensional problem of the scalar hyperbolic conservation law by the proposed method, we additionally use the iterative Godunov exact solver, because the Riemann invariants are non-conserved for moderate and strong shocks in an ideal gas. The proposed method belongs to a group of particle-in-cell (PIC) methods; to the best of the author's knowledge, there are no similar PIC numerical schemes using the Riemann invariants or the iterative Godunov exact solver. This article describes the application of the aforementioned method for the inviscid Burgers' equation, adiabatic gas dynamics equations, and the one-dimensional scalar hyperbolic conservation law. The numerical analysis results for several test cases (e.g., the standard shock-tube problem of Sod, the Riemann problem of Lax, the double expansion wave problem, the Shu–Osher shock-tube problem) are compared with the exact solution and Harten's data. In the shock for the proposed method, the flow properties change instantaneously (with an accuracy dependent on the grid cell size). The iterative Godunov exact solver determines the accuracy of the proposed method for flow discontinuities. In calculations, we use the iteration termination condition less than $10^{-5}$ to find the pressure difference between the current and previous iterations.

**Keywords:** gas; shock; Riemann problem; Godunov method; Lagrangian approach; numerical viscosity

## 1. Introduction

Simulating compressible flows is challenging owing to the presence of discontinuities in fluid flow, such as shocks, contact waves, and broad-band continuous flow scales. Numerous high-resolution schemes have been proposed to resolve smooth flow scales accurately and to capture the discontinuities simultaneously, e.g., the artificial viscosity method [1,2], total variation diminishing (TVD) method [3], essentially non-oscillatory (ENO) scheme [4], and weighted essentially non-oscillatory (WENO) scheme [5].

Harten et al. studied the ENO scheme [4,6–8], which was the first successful high-order method to enable the spatial discretization of hyperbolic conservation laws that had the ENO property. This property is considered to be very useful in the numerical simulation

of hyperbolic conservation laws, because high-order numerical methods often produce spurious oscillations—especially near shocks or other discontinuities. The finite-volume ENO spatial discretization has been studied [4], where it was shown to have uniform high-order accuracy in capturing the location of any discontinuity in fluid flow. Subsequently, Shu and Osher [9,10] developed the finite-difference ENO scheme. The main idea of the ENO scheme is to choose a stencil of interpolation points that suppresses oscillations, i.e., one chooses the stencil on which the solution varies the smoothest and approximates the flux at the cell boundaries with high-order accuracy, thus avoiding the large, spurious oscillations caused by interpolating data across discontinuities.

WENO schemes have been introduced [5,11,12] to address potential numerical instabilities when choosing ENO stencils. WENO methods use a convex combination of all of the ENO stencils; they achieve a higher order of accuracy than ENO methods in smooth regions, while retaining the ENO property at discontinuities.

The aforementioned methods follow the Eulerian approach. One disadvantage of these methods [1–24] is the presence of a numerical viscosity for shocks. Additionally, there are methods [25–28] using the exact Riemann solver; however, they too are Eulerian methods; moreover, these methods have numerical viscosity for shocks. Godunov et al. [25] proposed using moving grids to eliminate this disadvantage, but this dramatically complicates the problem, particularly in two- or three-dimensional cases. In this paper, we use a Lagrangian Godunov-type method and a fixed homogeneous grid to eliminate the disadvantage of numerical viscosity for shocks via a simplified approach. Additionally, we describe wall conditions. As discussed in the following sections, the proposed method has no numerical viscosity for shocks, as a well-known method of characteristics (MOC) [29–31]. In particular [30,31], the water hammer phenomenon modelling by the MOC using the Riemann invariants or solvers is considered. In the MOC, the convective terms are typically neglected, because the Riemann invariants in the water hammer problem are only weakly interdependent [30,31]. We use a similar approach to solve the one-dimensional adiabatic gas dynamics equations [29], but we split the equations into parts describing convection and acoustic processes separately. Different time steps are used to solve these equations because the local acoustic velocity (speed of sound) can differ many times from the convective velocity of the flow. When we are looking for the solution to the one-dimensional problem of the scalar hyperbolic conservation law via the proposed method, we additionally use the iterative Godunov exact solver [25], because the Riemann invariants are non-conserved for moderate and strong shocks. On the other hand, the proposed method is similar to particle-in-cell (PIC) methods. PIC methods are readily available in the literature [32–35]; to the best of the author's knowledge, there are no similar PIC numerical schemes using the Riemann invariants or the iterative Godunov exact solver. Finally, it should be mentioned that besides the Eulerian and Lagrangian approaches to the description of shocks, there is another approach based on the variational principle [36]; however, for this approach, when modeling flows, a numerical scheme of Eulerian or Lagrangian type is still needed.

## 2. Materials and Methods

### 2.1. A Methodological Concept for the Inviscid Burgers' Equation

For a better understanding of an idea of the proposed method, we describe an algorithm of the method for the inviscid Burgers' equation [37]. The inviscid Burgers' equation is a scalar nonlinear equation given as follows:

$$\frac{\partial u}{\partial t} + u \frac{\partial u}{\partial x} = 0, \tag{1}$$

where u is the convective velocity of flow, x is the direction, and t is the time. Equation (1) can be solved using the proposed Lagrangian scheme. The cells are considered in pairs. We

will use two fundamental solutions of the inviscid Burgers' equation: shock, and rarefaction wave. The cells' coordinates after transferring are determined as follows:

$$x_1 = x_i + u_i^{k-1} \Delta t,$$
$$x_2 = x_{i+1} + u_{i+1}^{k-1} \Delta t,$$

(2)

where $\Delta t$ is the time step. Furthermore, we use the following parameters: $e_1 = 10^{-4}$, which is the tolerance associated with the comparison of the flow variables; and $e_2 = 10^{-8}$, which is the machine arithmetic tolerance associated with the comparison of the x coordinates.

The solution is sought in the following form:

(1)　If the conditions

$$|u_{i+1} - u_i| < e_1,$$

(3)

are satisfied, then for all grid cells for which the conditions are satisfied, the solution at the next time moment k is trivial:

$$u_j^k = \frac{1}{2}(u_i^{k-1} + u_{i+1}^{k-1}).$$

(4)

(2)　If the conditions

$$(u_{i+1} - u_i) < -e_1$$

(5)

are satisfied, then we have the shock.

　　(2.1) if

$$(x_2 - x_1) > e_2,$$

(6)

then the solution is defined as

$$(2.1\,a)\ \text{if}\ \ x_1 \leq x_j \leq (x_1 + x_2)/2\ \ \text{then}\ \ u_j^k = u_i^{k-1},$$
$$(2.1\,b)\ \text{if}\ \ (x_1 + x_2)/2 < x_j \leq x_2\ \ \text{then}\ \ u_j^k = u_{i+1}^{k-1}.$$

(7)

　　(2.2) If

$$(x_2 - x_1) \leq e_2,$$

(8)

then the discontinuity boundary is first determined as

$$x = x_i + h/2 + 0.5(u_i^{k-1} + u_{i+1}^{k-1}) \Delta t,$$

(9)

and the solution is assigned to cell j to the left of the discontinuity boundary

$$u_j^k = u_i^{k-1},$$

(10)

and in cell $j + 1$ to the right of the discontinuity boundary, the following values are assigned:

$$u_j^k = u_{i+1}^{k-1}.$$

(11)

(3)　If

$$(u_{i+1} - u_i) \geq e_1,$$

(12)

then we have the rarefaction wave, and the solution for $x_1 \leq x_j \leq x_2$ is determined as follows:

$$u_j^k = u_i^{k-1} + \frac{u_{i+1}^{k-1} - u_i^{k-1}}{x_2 - x_1}(x_j - x_1).$$

(13)

In this case, the loop over all $x_i$ is performed twice: first, for all $u_i \geq e_1$ from $n-1$ to 1, and second, for all $u_i \leq -e_1$ from 1 to $n-1$. Here, n is the number of grid cells. When conditions (1)–(3) are satisfied, if it turns out that the solution has already been assigned to the grid cell, then it is replaced by a new one. When assigning the solution to the grid cells, the condition is separately checked so that the propagation of this solution at the convection velocity does not overtake (does not overwrite) the solution of the forward shock, if one exists. For this, at the beginning, the positions of all of the shock boundaries at the next moment in time are determined.

The modeling area is taken to be $x \in [a, b]$. The grid contains n cells, and the grid step h is

$$h = \frac{b-a}{n},$$ (14)

as shown in Figure 1. The time step was chosen similar to the Courant criterion.

$$\Delta t = k_u \frac{h}{u_s},$$ (15)

where $u_s$ is the convective velocity of the shock, and $k_u$ is an integer. For example, if $k_u = 4$, then the shock propagates to the four grid cells.

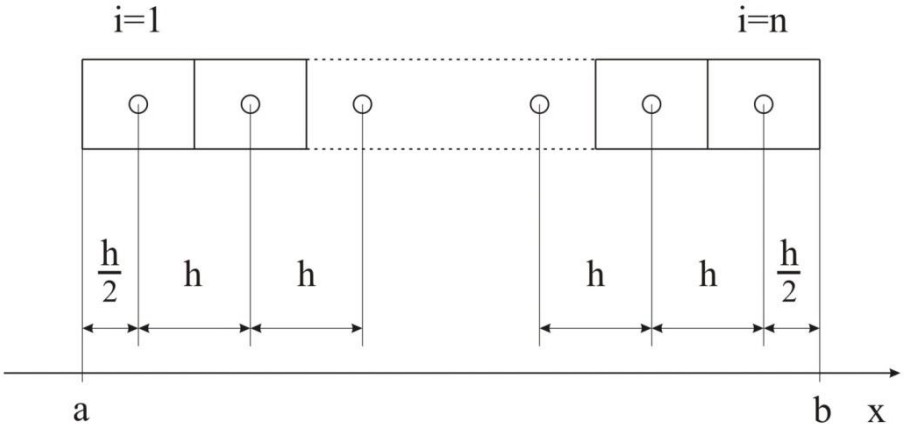

**Figure 1.** The modeling area and the grid of the proposed method.

## 2.2. Basic Terms and a Methodological Concept for Adiabatic Gas Dynamics Equations

Consider the one-dimensional adiabatic gas dynamics equations [37]:

$$\frac{\partial \rho}{\partial t} + u \frac{\partial \rho}{\partial x} + \rho \frac{\partial u}{\partial x} = 0,$$
$$\frac{\partial u}{\partial t} + u \frac{\partial u}{\partial x} + \frac{1}{\rho} \frac{\partial p}{\partial x} = 0.$$ (16)

Here, $\rho$ is the density and p is the pressure.
The pressure is related to the adiabatic law

$$\frac{p}{\rho^\kappa} = \frac{p_0}{\rho_0^\kappa} = A = \text{const},$$ (17)

where $\kappa$ is the ratio of specific heat coefficients at constant pressure and volume. For air, $\kappa = 1.4$. Values $\rho_0$ and $p_0$ can be chosen from initial conditions of a considered problem.

As previously shown [38], the system of Equation (16) can be represented as a system of wave equations as follows:

$$\frac{\partial w_1}{\partial t} + (u - c) \frac{\partial w_1}{\partial x} = 0,$$
$$\frac{\partial w_2}{\partial t} + (u + c) \frac{\partial w_2}{\partial x} = 0,$$ (18)

where c is the sound velocity, and $w_i$ represents variables that are determined as follows:

$$w_1 = u - \widetilde{P},$$
$$w_2 = u + \widetilde{P}.$$

(19)

Here, the function $\widetilde{P}$ is

$$\widetilde{P} = \frac{2c_0}{\kappa - 1}\left(\left(\frac{\rho}{\rho_0}\right)^{\frac{\kappa-1}{2}} - 1\right),$$

(20)

where $c_0$ is the sound velocity, which is determined from the initial conditions as follows:

$$c_0 = \sqrt{\kappa\frac{P_0}{\rho_0}}.$$

(21)

Equation (18) describes a movement of two waves with velocities $u - c$ and $u + c$, and their respective transfer values $w_1$ and $w_2$. The system of Equation (18) can be solved using the Lagrangian approach. In this paper, the following method is proposed: The stages of convection and acoustics are considered separately, and splitting into physical processes is performed because the local acoustic velocity (speed of sound) can differ greatly from the convective velocity of the flow. Thus, at the acoustic stage we solve the system

$$\frac{\partial \rho}{\partial t} + \rho\frac{\partial u}{\partial x} = 0,$$
$$\frac{\partial u}{\partial t} + \frac{1}{\rho}\frac{\partial P}{\partial x} = 0,$$

(22)

obtained from (16) by discarding the convective terms; its corresponding system of wave equations is

$$\frac{\partial w_1}{\partial t} - c\frac{\partial w_1}{\partial x} = 0,$$
$$\frac{\partial w_2}{\partial t} + c\frac{\partial w_2}{\partial x} = 0,$$

(23)

The system (23) is obtained from (22) with help of (19).

At the convection stage, we solve the system with the convective terms only:

$$\frac{\partial \rho}{\partial t} + u\frac{\partial \rho}{\partial x} = 0,$$
$$\frac{\partial u}{\partial t} + u\frac{\partial u}{\partial x} = 0.$$

(24)

We apply the numerical scheme principle described above for the inviscid Burgers' equation to find the solution of the systems (23) and (24); for more details, see Appendix A.

The time step at the acoustics stage was chosen according to the Courant criterion.

$$\Delta t_c = k_c\frac{h}{c_s},$$

(25)

where $c_s$ is the acoustic velocity of the shock. For example, if $k_c = 3$, then the shock propagates through three grid cells at the acoustics stage.

The time step at the convection stage was chosen similar to that of (25):

$$\Delta t_u = k_u\frac{h}{u_s},$$

(26)

where $u_s$ is the convective velocity of the shock. For example, if $k_u = 2$, then the shock propagates to two grid cells at the convection stage. If $_\Delta t_c \neq {}_\Delta t_u$, then the marching time step $_\Delta t$ is chosen so that

$$_\Delta t_c = n_{c\Delta} t,$$
$$_\Delta t_u = n_{u\Delta} t, \tag{27}$$

where $n_c$ and $n_u$ are integers.

*2.3. Basic Terms and a Methodological Concept for the One-Dimensional Scalar Hyperbolic Conservation Law*

Consider the one-dimensional scalar hyperbolic conservation law [13]:

$$\frac{\partial \rho}{\partial t} + u\frac{\partial \rho}{\partial x} + \rho\frac{\partial u}{\partial x} = 0,$$
$$\frac{\partial u}{\partial t} + u\frac{\partial u}{\partial x} + \frac{1}{\rho}\frac{\partial p}{\partial x} = 0, \tag{28}$$
$$\frac{\partial \varepsilon}{\partial t} + u\frac{\partial \varepsilon}{\partial x} + \frac{p}{\rho}\frac{\partial u}{\partial x} = 0.$$

Here, $\varepsilon$ is the internal specific energy.

The pressure is related to thermodynamic state variables $\rho$ and $\varepsilon$ through the following state equation:

$$p = (\kappa - 1)\,\rho\,\varepsilon. \tag{29}$$

Godunov et al. [25] proposed the iterative exact Riemann solver implemented in a finite-difference scheme. Instead of this method, other researchers later used a control volume method, while we used the Lagrangian approach. In this paper, the following method is proposed: The stages of convection and acoustics are considered separately, and splitting into physical processes is performed because the local acoustic velocity (speed of sound) can differ greatly from the convective velocity of the flow. Thus, at the acoustic stage we solve the system

$$\frac{\partial \rho}{\partial t} + \rho\frac{\partial u}{\partial x} = 0,$$
$$\frac{\partial u}{\partial t} + \frac{1}{\rho}\frac{\partial p}{\partial x} = 0, \tag{30}$$
$$\frac{\partial \varepsilon}{\partial t} + \frac{p}{\rho}\frac{\partial u}{\partial x} = 0.$$

and the Equation (29). The system (30) is obtained from (28) by discarding the convective terms, and it can be represented in a matrix form as follows:

$$\left\{ \begin{array}{c} \frac{\partial \rho}{\partial t} \\ \frac{\partial u}{\partial t} \\ \frac{\partial \varepsilon}{\partial t} \end{array} \right\} = [A] \left\{ \begin{array}{c} \frac{\partial \rho}{\partial x} \\ \frac{\partial u}{\partial x} \\ \frac{\partial \varepsilon}{\partial x} \end{array} \right\} \tag{31}$$

where $[A]$ is the Jacobian matrix

$$[A] = \begin{bmatrix} 0 & \rho & 0 \\ \frac{(\kappa-1)\varepsilon}{\rho} & 0 & \kappa - 1 \\ 0 & (\kappa-1)\varepsilon & 0 \end{bmatrix}. \tag{32}$$

The matrix $[A]$ can be expressed as

$$[A] = [R][\Lambda][L] \tag{33}$$

in terms of a diagonal matrix $[\Lambda]$ of eigenvalues of the Jacobian matrix

$$[\Lambda] = \begin{bmatrix} -c & 0 & 0 \\ 0 & 0 & 0 \\ 0 & 0 & c \end{bmatrix}, \tag{34}$$

a matrix of corresponding left eigenvectors $[L]$, determined as

$$[L] = \begin{bmatrix} -\frac{c}{\kappa\rho} & 1 & -\frac{\kappa-1}{c} \\ 1 & 0 & -\frac{\rho\kappa}{c^2} \\ \frac{c}{\kappa\rho} & 1 & \frac{\kappa-1}{c} \end{bmatrix}, \tag{35}$$

and a matrix of corresponding right eigenvectors $[R]$, defined as

$$[R] = \begin{bmatrix} -\frac{\rho}{2c} & \frac{\kappa-1}{\kappa} & \frac{\rho}{2c} \\ \frac{1}{2} & 0 & \frac{1}{2} \\ -\frac{c}{2\kappa} & -\frac{(\kappa-1)\varepsilon}{\kappa\rho} & \frac{c}{2\kappa} \end{bmatrix}. \tag{36}$$

Since

$$[L][R] = [E], \tag{37}$$

where $[E]$ is a diagonal identity matrix, the multiplication of Equation (31) from the left by $[L]$ and the use of (37) gives

$$[L] \left\{ \begin{array}{c} \frac{\partial \rho}{\partial t} \\ \frac{\partial u}{\partial t} \\ \frac{\partial \varepsilon}{\partial t} \end{array} \right\} = [\Lambda][L] \left\{ \begin{array}{c} \frac{\partial \rho}{\partial x} \\ \frac{\partial u}{\partial x} \\ \frac{\partial \varepsilon}{\partial x} \end{array} \right\}. \tag{38}$$

If $[L] = [\mathrm{const}]$, we can obtain from (38) the characteristic form

$$\frac{\partial w_1}{\partial t} - c\frac{\partial w_1}{\partial x} = 0,$$
$$\frac{\partial w_2}{\partial t} = 0, \tag{39}$$
$$\frac{\partial w_3}{\partial t} + c\frac{\partial w_3}{\partial x} = 0,$$

where

$$w_1 = -\frac{c*}{\kappa\rho*}\rho + u - \frac{\kappa-1}{c*}\varepsilon,$$
$$w_2 = \rho - \frac{\kappa\rho*}{c*^2}\varepsilon, \tag{40}$$
$$w_3 = \frac{c*}{\kappa\rho*}\rho + u + \frac{\kappa-1}{c*}\varepsilon.$$

Here, an asterisk indicates constant values. Equation (39) describes the movement of two waves with velocities $-c$ and $c$, and their respective transfer values, $w_1$ and $w_3$ (40).

The system of Equation (39) can be solved using the proposed Lagrangian approach. However, in this paper, we use the Godunov exact solver for the solution of the full nonlinear system (28), (29), and then subtract the convective velocity from the solution to obtain the solution of the Equation (30). For more details, see Appendix B.

At the convection stage, we solve the system with the convective terms only:

$$\frac{\partial \rho}{\partial t} + u\frac{\partial \rho}{\partial x} = 0,$$
$$\frac{\partial u}{\partial t} + u\frac{\partial u}{\partial x} = 0, \tag{41}$$
$$\frac{\partial \varepsilon}{\partial t} + u\frac{\partial \varepsilon}{\partial x} = 0,$$

as well as Equation (29).

The time step $_\Delta t_c$ at the acoustics stage was chosen according to the Courant criterion (25). The time step $_\Delta t_u$ at the convection stage was chosen as (26).

## 3. Results

In this section, we apply the proposed method to solve several problems with different initial and boundary conditions.

### 3.1. Results for the Inviscid Burgers' Equation

Firstly, we considered the initial value problem [39] for the inviscid Burgers' equation with the initial conditions

$$
\begin{aligned}
u_j^0 &= 0, \text{ if } x_j < 0, \\
u_j^0 &= 1, \text{ if } 0 \le x_j \le 1, \\
u_j^0 &= 0, \text{ if } x_j > 1.
\end{aligned}
\tag{42}
$$

The exact solution of the shock–rarefaction wave problem (42) was determined as follows:

$$
u(t, x) =
\begin{cases}
0, & \text{if } x < 0 \\
\frac{x}{t}, & \text{if } 0 \le x \le t \\
1, & \text{if } t < x \le \frac{t}{2} + 1 \\
0, & \text{if } x > \frac{t}{2} + 1
\end{cases}
\quad , \text{ for } t \le 2,
\tag{43}
$$

and

$$
u(t, x) =
\begin{cases}
0, & \text{if } x < 0 \\
\frac{x}{t}, & \text{if } 0 \le x \le \sqrt{2t} \\
0, & \text{if } x > \sqrt{2t}
\end{cases}
\quad , \text{ for } t > 2.
\tag{44}
$$

The modeling area was taken to be $x \in [-1, \, 4]$. The grid contained 50 cells, and the grid step was $h = 0.1$. The time step was chosen according to the Courant criterion (15), where $u_s = 0.5$ is the convective velocity of the shock in (42), where $k_u = 4$, i.e., the shock propagates to the four grid cells. The time step was $_\Delta t = 0.8$. It should be noted that the rarefaction wave (at its right boundary) has the convective velocity $u_r = 1.0$ in (43), and propagates to the eight grid cells. At the time $t = 2$, the rarefaction wave catches up with the shock, but is not able to overtake it (44).

The results after two and five time steps are shown in Figures 2 and 3 in comparison to the exact solutions (43) and (44), respectively. We can see the good agreement between the numerical and exact solutions, and that the proposed method does not have numerical viscosity for shocks. However, owing to the accumulation of the roundoff error, the shock in Figure 3 overtakes the exact solution by one grid cell at the shown moment in time.

Secondly, we considered the well-known test problem [40] with the periodic initial function

$$
u_j^0 = 0.5 \sin(-\pi x).
\tag{45}
$$

The modeling area was taken to be $x \in [-1, \, 1]$. The grid contained 80 cells, and the grid step was $h = 0.025$. The time step was chosen according to the Courant criterion (15), where $u_s = 0.5$ is the convective velocity of the shock in (45), where $k_u = 2$, i.e., the shock propagates to the two grid cells. The time step was $_\Delta t = 0.1$.

The result after 11 time steps is shown in Figure 4 in comparison to the exact solution [40]. We can see the good agreement between numerical and exact solutions, and that the proposed method does not have numerical viscosity for shocks.

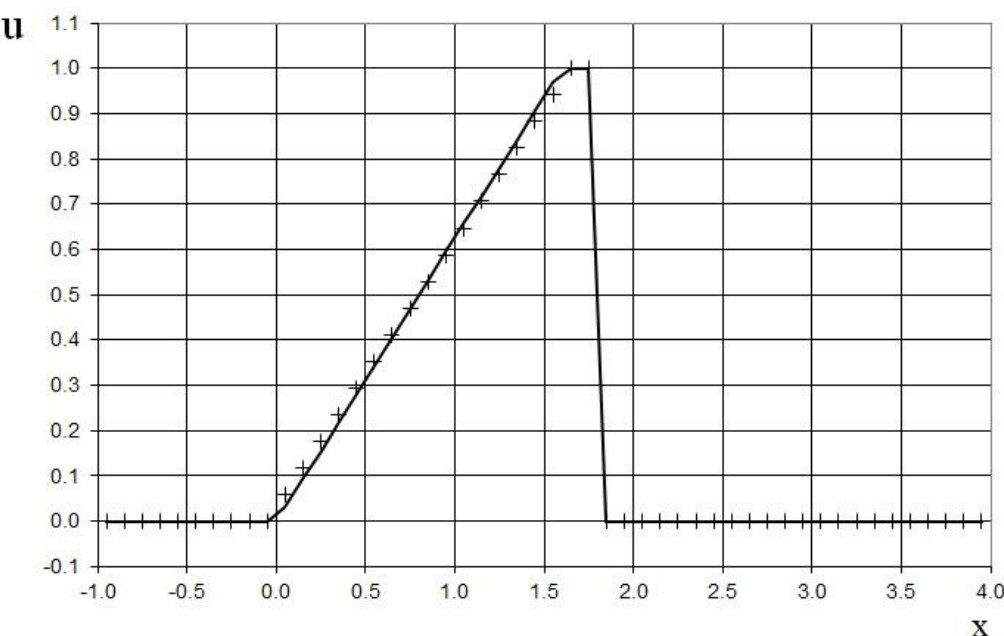

**Figure 2.** Velocity plot for the shock–rarefaction wave problem for the inviscid Burgers' equation at the time t = 1.6: the exact solution (line), and the present method (cross).

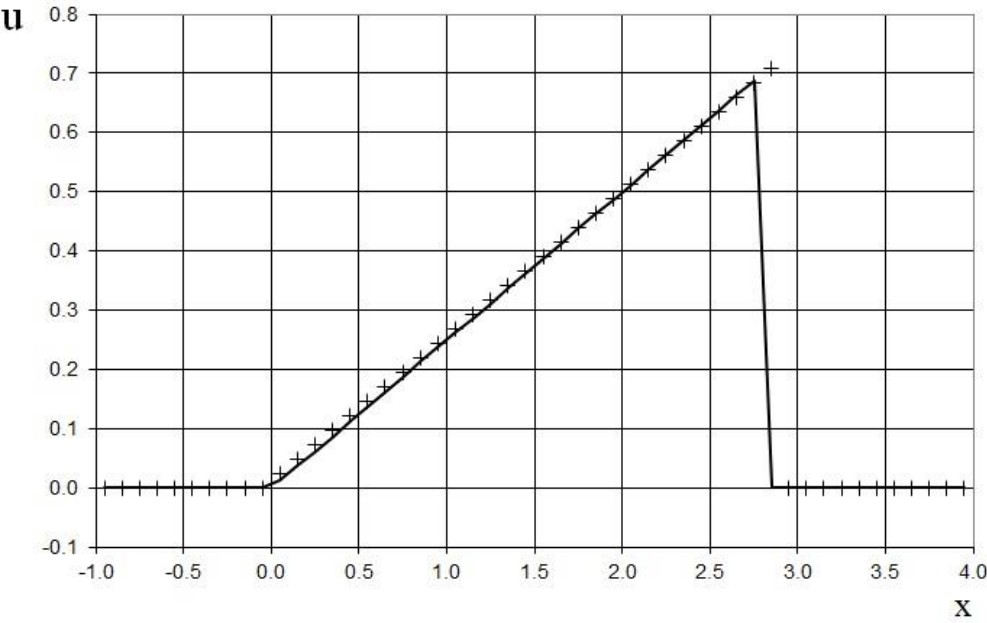

**Figure 3.** Velocity plot for the shock–rarefaction wave problem for the inviscid Burgers' equation at the time t = 4.0: the exact solution (line), and the present method (cross).

*3.2. Results for the Adiabatic Gas Dynamics Equations*

The first example for the adiabatic gas dynamics equations is the standard shock-tube problem with the following initial conditions:

$$p_j^0 = 1,\ u_j^0 = 0,\ \rho_j^0 = 1,\ \text{if } x_j < 0,$$

$$p_j^0 = 0.378929142,\ u_j^0 = 0,\ \rho_j^0 = 0.5,\ \text{if } x_j \geq 0. \tag{46}$$

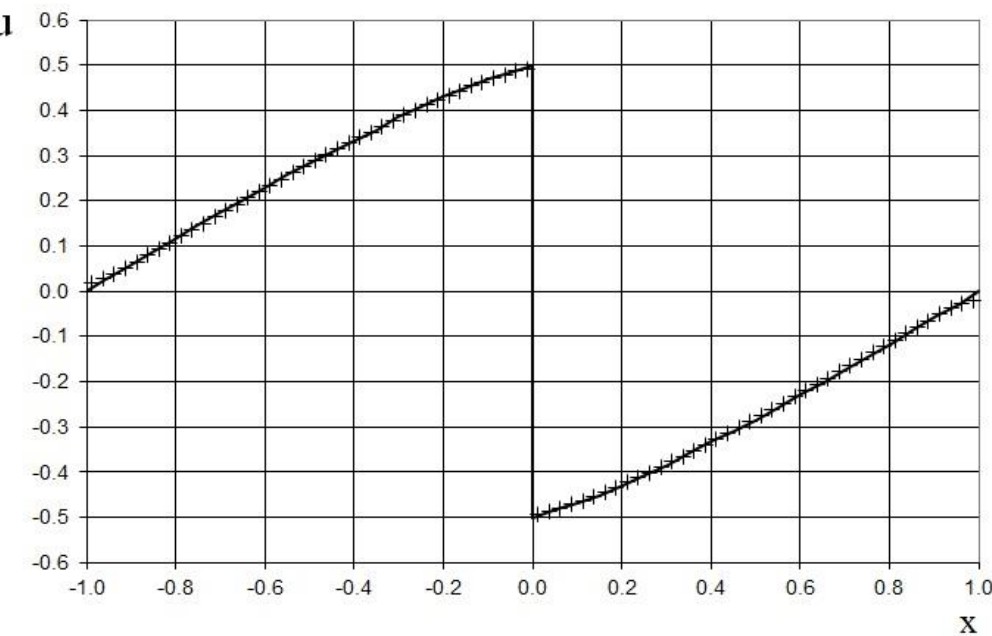

**Figure 4.** Velocity plot for the problem (45) for the inviscid Burgers' equation (t = 1.1): the exact solution [40] (line), and the present method (cross).

The modeling area was taken to be x ∈ [−1, 1]. The grid contained 200 cells, and the grid step was h = 0.01. The time step at the acoustics stage $\Delta t_c$ = 0.054218535053 was chosen according to the Courant criterion (25), where $k_c$ = 6, i.e., the shock propagates through six grid cells at the acoustics stage. The time step at the convection stage $\Delta t_u$ = 0.052230695168 was chosen according to a similar criterion (26), where $k_u$ = 2, i.e., the shock propagates to two grid cells at the convection stage. The marching time step was chosen as $\Delta t =_\Delta t_c =_\Delta t_u$ = 0.052230695168.

The results after 10 marching time steps t = 0.52230695168 are shown in Figures 5–7 in comparison to the exact solution. Figures 5–7 show the absence of the numerical viscosity for the shock. However, owing to the accumulation of the roundoff error, the shock overtakes the exact solution by two grid cells at the shown moment in time.

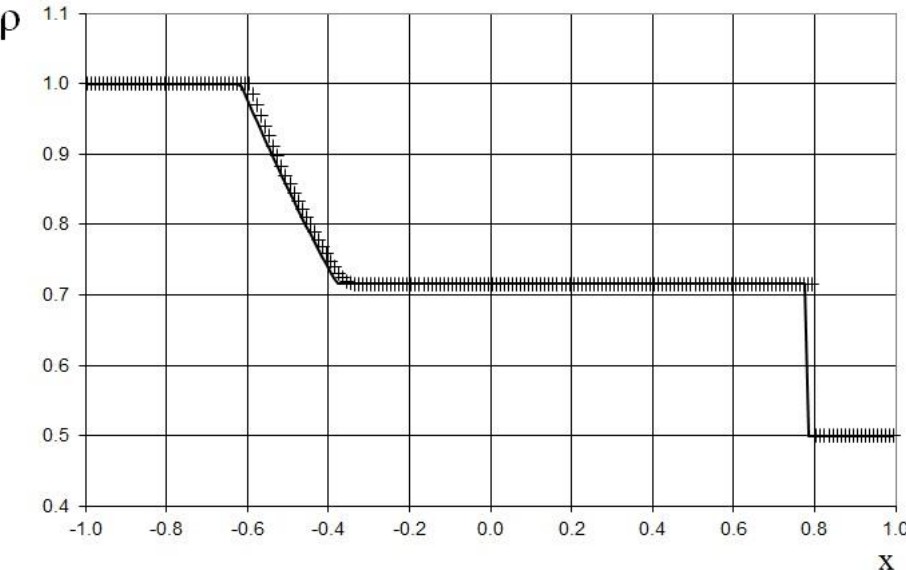

**Figure 5.** Density plot for the shock-tube problem (46) for the adiabatic gas dynamics equations: the exact solution (line), and the present method (cross).

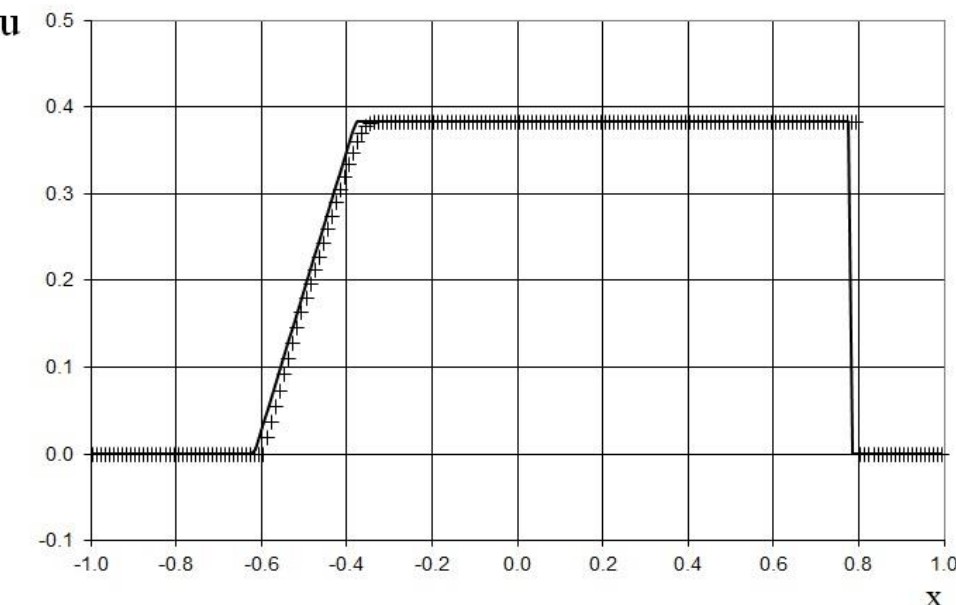

**Figure 6.** Velocity plot for the shock-tube problem (46) for the adiabatic gas dynamics equations: the exact solution (line), and the present method (cross).

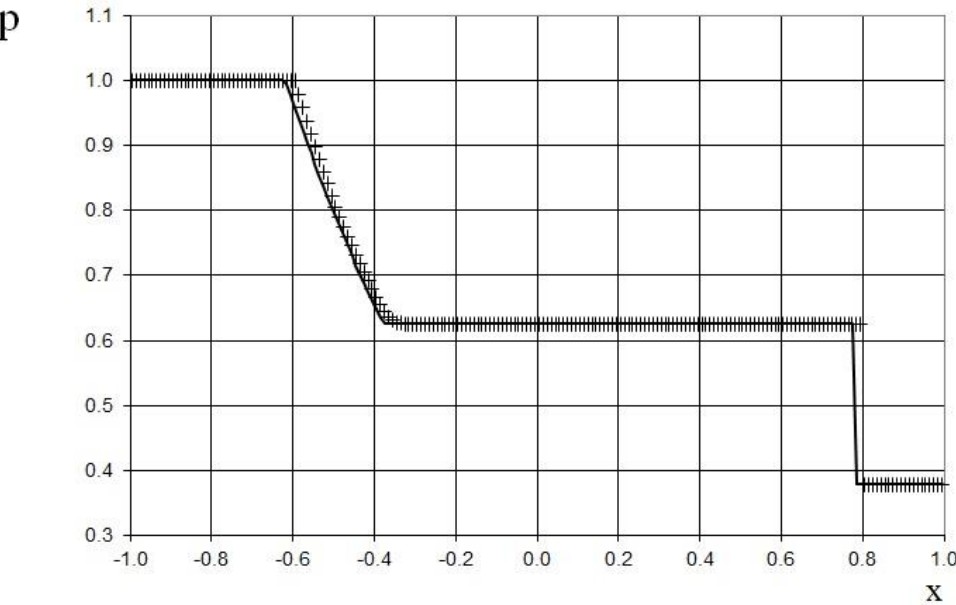

**Figure 7.** Pressure plot for the shock-tube problem (46) for the adiabatic gas dynamics equations: the exact solution (line), and the present method (cross).

The second example for the adiabatic gas dynamics equations is the double expansion wave problem [41] with the following initial conditions:

$$p_j^0 = 10^5, \ u_j^0 = -100, \ \rho_j^0 = 1.2, \ \text{if } x_j < 0,$$
$$p_j^0 = 10^5, \ u_j^0 = 100, \ \rho_j^0 = 1.2, \ \text{if } x_j \geq 0. \tag{47}$$

The modeling area was taken to be $x \in [-0.5, 0.5]$. The grid contained 100 cells, and the grid step was $h = 0.01$. The time step at the acoustics stage $_\Delta t_u = 0.000204939017$ was chosen according to the Courant criterion (25), where $k_c = 7$, i.e., the shock propagates through seven grid cells at the acoustics stage. The time step at the convection stage $_\Delta t_u = 0.0002$ was chosen according to a similar criterion (26), where $k_u = 2$, i.e., the shock

propagates to two grid cells at the convection stage. The marching time step was chosen as $\Delta t =_\Delta t_c =_\Delta t_u = 0.0002$.

The results after four marching time steps $t = 0.0008$ are shown in Figures 8–10 in comparison to the exact solution. Figures 8–10 show a good agreement between the exact and numerical solutions for the double expansion wave problem.

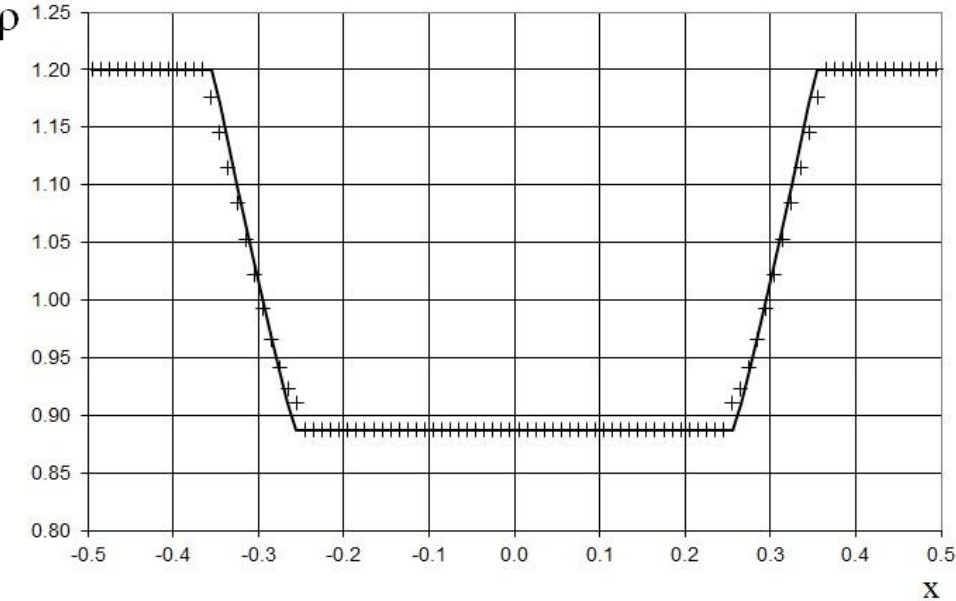

**Figure 8.** Density plot for the double expansion wave problem (47) for the adiabatic gas dynamics equations: the exact solution (line), and the present method (cross).

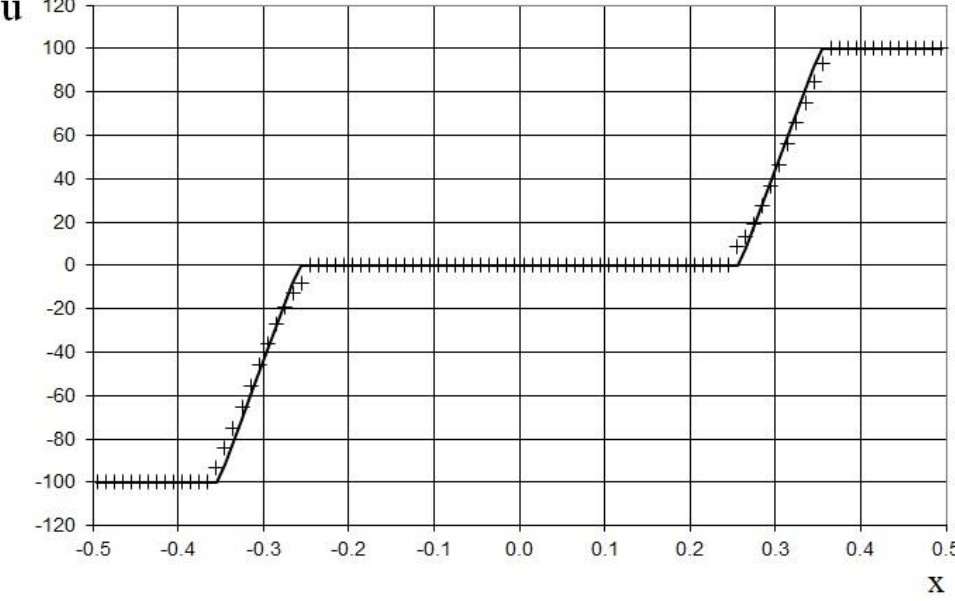

**Figure 9.** Velocity plot for the double expansion wave problem (47) for the adiabatic gas dynamics equations: the exact solution (line), and the present method (cross).

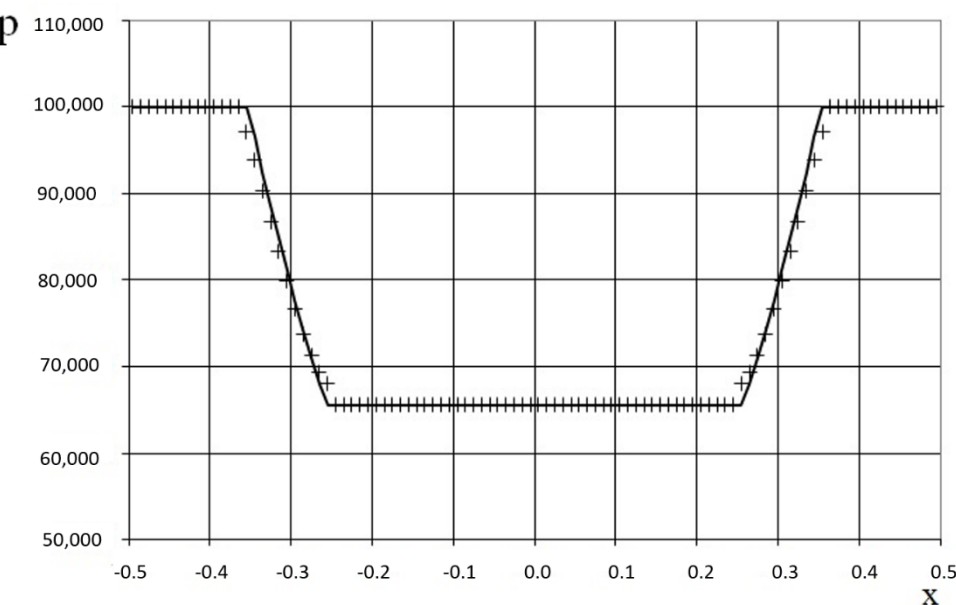

**Figure 10.** Pressure plot for the double expansion wave problem (47) for the adiabatic gas dynamics equations: the exact solution (line), and the present method (cross).

*3.3. Results for the One-Dimensional Scalar Hyperbolic Conservation Law*

The first example is the standard shock-tube problem of Sod [42], with the following initial conditions:

$$p_j^0 = 1, \ u_j^0 = 0, \ \rho_j^0 = 1, \ \text{if } x_j < 0,$$
$$p_j^0 = 0.1, \ u_j^0 = 0, \ \rho_j^0 = 0.125, \ \text{if } x_j \geq 0.$$
(48)

The modeling area was taken to be $x \in [-4.5, 5.5]$. The grid contained 100 cells, and the grid step was $h = 0.1$. The time step at the acoustics stage was chosen according to the Courant criterion (25), where $k_c = 1$, i.e., the shock propagates through one grid cell at the acoustics stage. The time step at the convection stage was chosen according to (26), where $k_u = 2$, i.e., the shock propagates to two grid cells at the convection stage. The marching time step was $_\Delta t_u = 0.02020929$. The acoustics stage was performed in 6 time steps $_\Delta t_c = 6_\Delta t$, and the convection stage was performed after 10 steps $_\Delta t_u = 10_\Delta t$.

The results after 110 marching time steps are shown in Figures 11–13 in comparison to the exact solution and the data obtained by Harten [3] via the ULT1C scheme. Note that more modern numerical schemes, such as WENO [14,24], have even higher numerical viscosity for shocks. Therefore, we chose the ULT1C scheme for comparison. Figures 11–13 show the absence of the numerical viscosity for the shock, in contrast to the Harten method. However, owing to the accumulation of the roundoff error, the shock is delayed by one grid cell at the shown moment in time.

The second example is the Riemann problem of Lax [43], with the following initial conditions:

$$p_j^0 = 3.52773, \ u_j^0 = 0.69888, \ \rho_j^0 = 0.445, \ \text{if } x_j < 0,$$
$$p_j^0 = 0.571, \ u_j^0 = 0, \ \rho_j^0 = 0.5, \ \text{if } x_j \geq 0.$$
(49)

The modeling area was taken as $x \in [-8, 6]$. The grid contained 140 cells, and the grid step was $h = 0.1$. The time step at the acoustics stage was chosen according to the Courant criterion (25), where $k_c = 2$, i.e., the shock propagates through two grid cells at the acoustics stage. The time step at the convection stage was chosen according to (26), where $k_u = 3$, i.e., the shock propagates through three grid cells at the convection stage. The marching time step was $_\Delta t = 0.2$. The acoustics and convection stages were performed at each marching time step, i.e., $_\Delta t_c = _\Delta t_u = _\Delta t$.

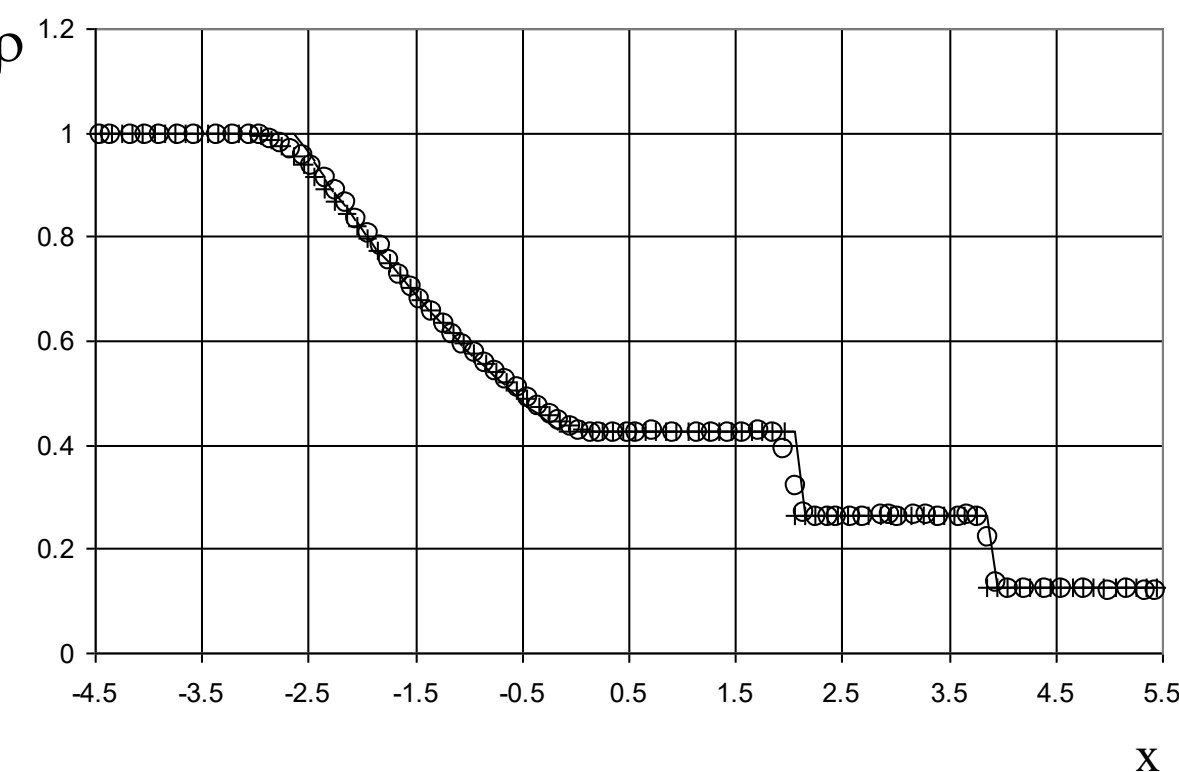

**Figure 11.** Density plot for the shock-tube problem of Sod: the exact solution (line), the ULT1C scheme of Harten (circle), and the present method (cross).

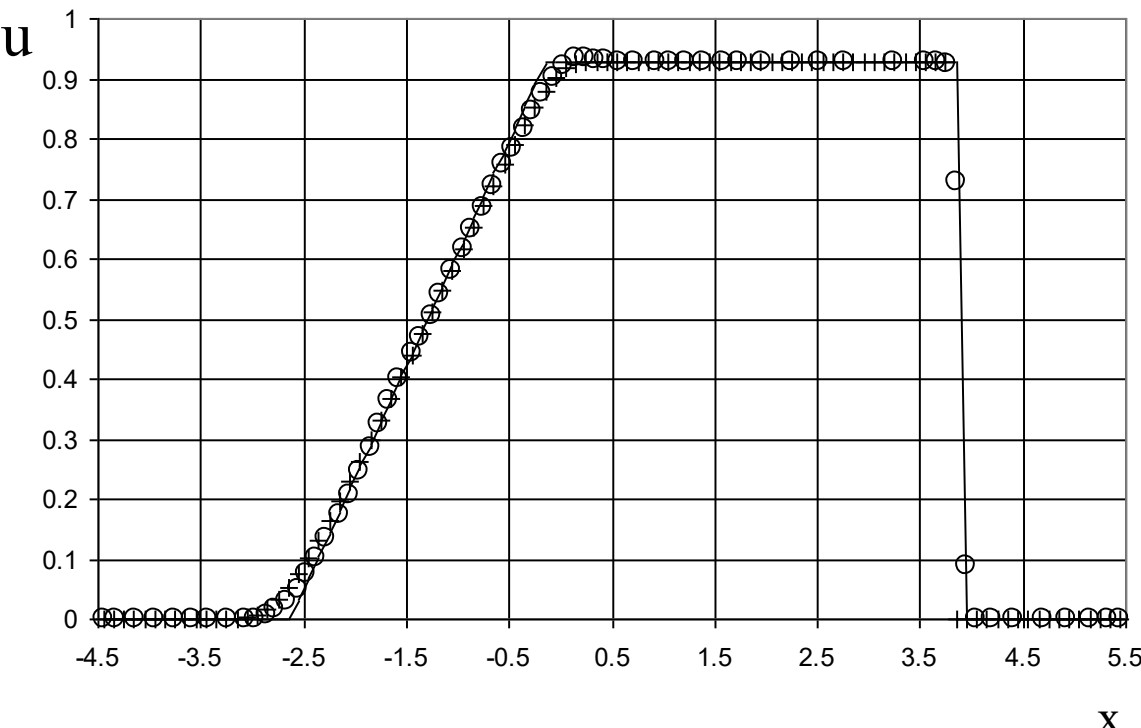

**Figure 12.** Velocity plot for the shock-tube problem of Sod: the exact solution (line), the ULT1C scheme of Harten (circle), and the present method (cross).

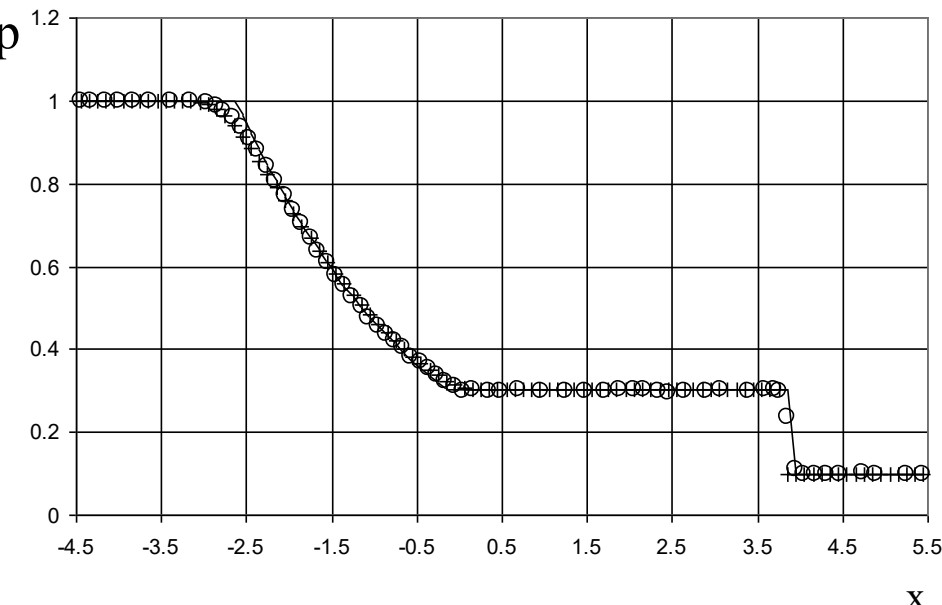

**Figure 13.** Pressure plot for the shock-tube problem of Sod: the exact solution (line), the ULT1C scheme of Harten (circle), and the present method (cross).

The results are shown in Figures 14–16 in comparison with the exact solution and the data obtained by Harten [3] using the ULT1C scheme at the time t = 2.0. Figures 14–16 show the absence of the numerical viscosity for the shock, in contrast to the Harten method. However, owing to the accumulation of the roundoff error, the shear wave is delayed by one grid cell at the shown moment in time.

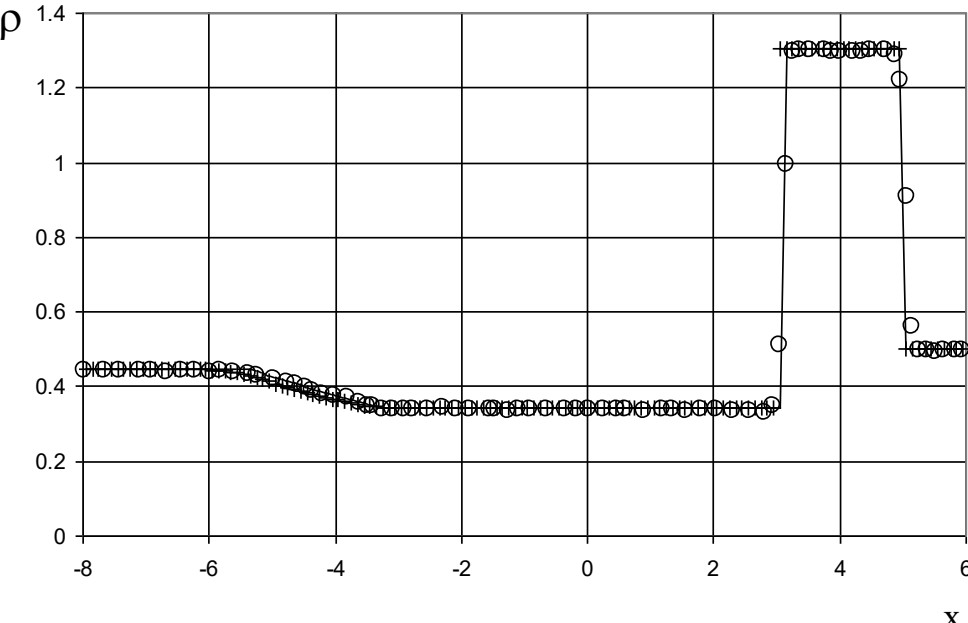

**Figure 14.** Density plot for the Riemann problem of Lax: the exact solution (line), the ULT1C scheme of Harten (circle), and the present method (cross).

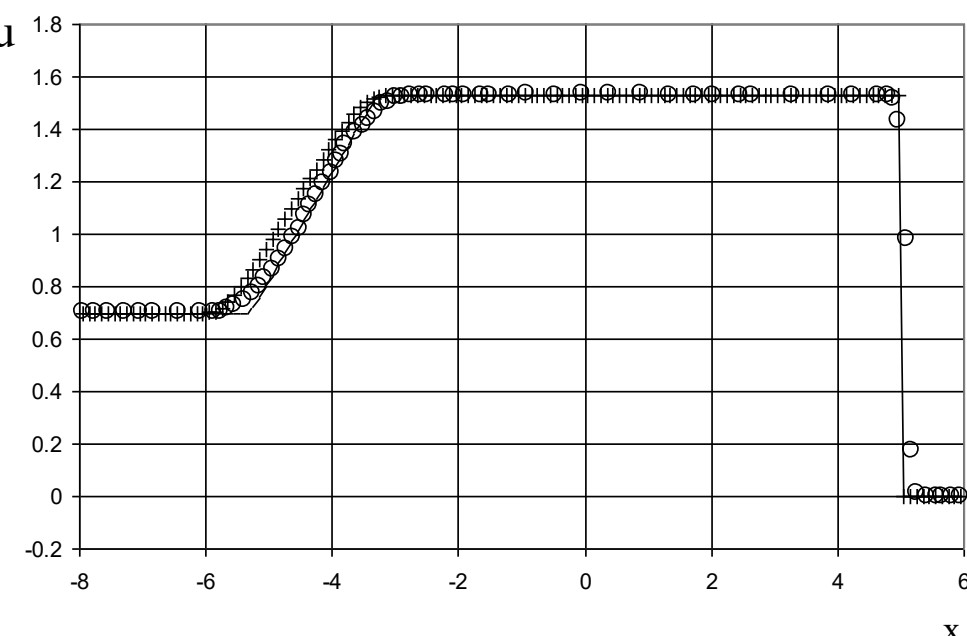

**Figure 15.** Velocity plot for the Riemann problem of Lax: the exact solution (line), the ULT1C scheme of Harten (circle), and the present method (cross).

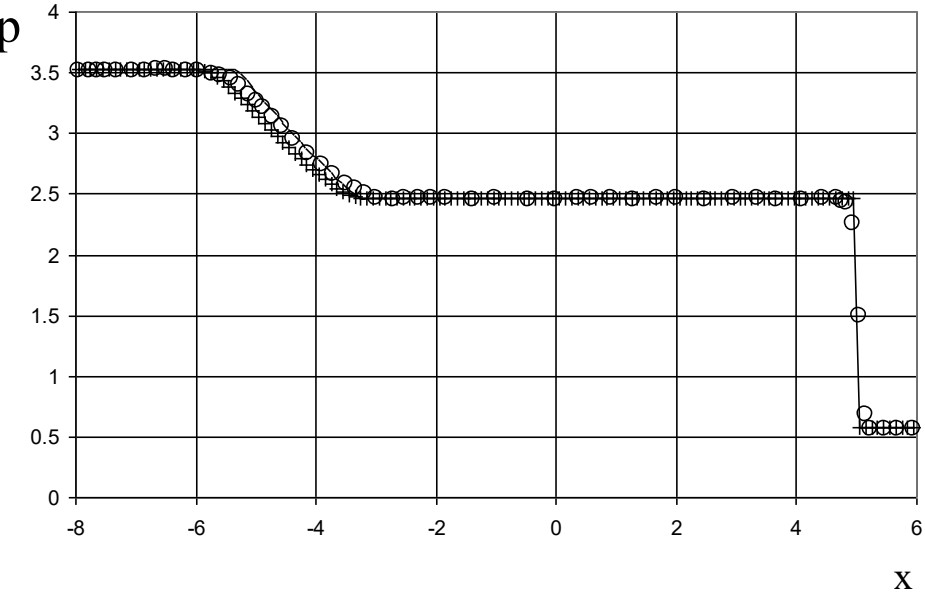

**Figure 16.** Pressure plot for the Riemann problem of Lax: the exact solution (line), the ULT1C scheme of Harten (circle), and the present method (cross).

The third example for the one-dimensional scalar hyperbolic conservation law is the double expansion wave problem [41] with the initial conditions given by (47).

The modeling area was taken to be $x \in [-0.5, 0.5]$. Two kinds of grids were used: a coarse grid (100 grid points), and a finer grid (1000 grid points); the grid step was $h = 0.01$ and $h = 0.001$, respectively. The time step at the acoustics stage was chosen according to the Courant criterion (25), where $k_c = 7$—i.e., the shock propagates through seven grid cells at the acoustics stage—for both the coarse grid and the finer grid. The time step at the convection stage was chosen according to (26), where $k_u = 2$—i.e., the shock propagates through two grid cells at the convection stage—for both grids. The marching time step was $\Delta t = 0.0002$ for the coarse grid and $\Delta t = 0.00002$ for the finer grid. The acoustics and convection stages were performed at each marching time step, i.e., $\Delta t_c = \Delta t_u = \Delta t$.

The results after 4 marching time steps for the coarse grid and after 40 marching time steps for the finer grid (t = 0.0008) are shown in Figures 17–22 in comparison to the exact solution. Figures 17–19 show a good agreement and Figures 20–22 show a better agreement between the exact and numerical solutions for the double expansion wave problem.

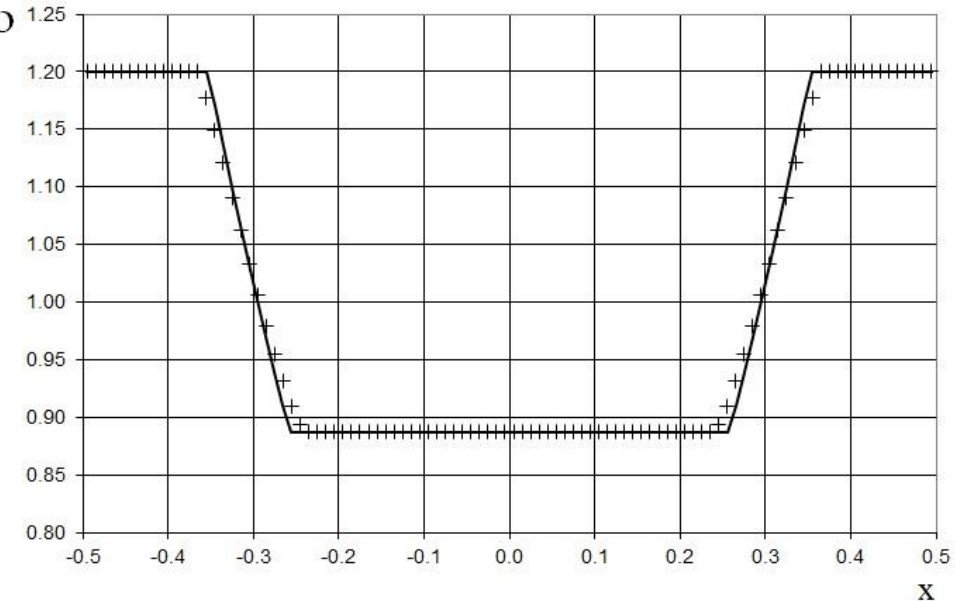

**Figure 17.** Density plot for the double expansion wave problem (47) for the one-dimensional scalar hyperbolic conservation law, using the coarse grid (100 grid points): the exact solution (line), and the present method (cross).

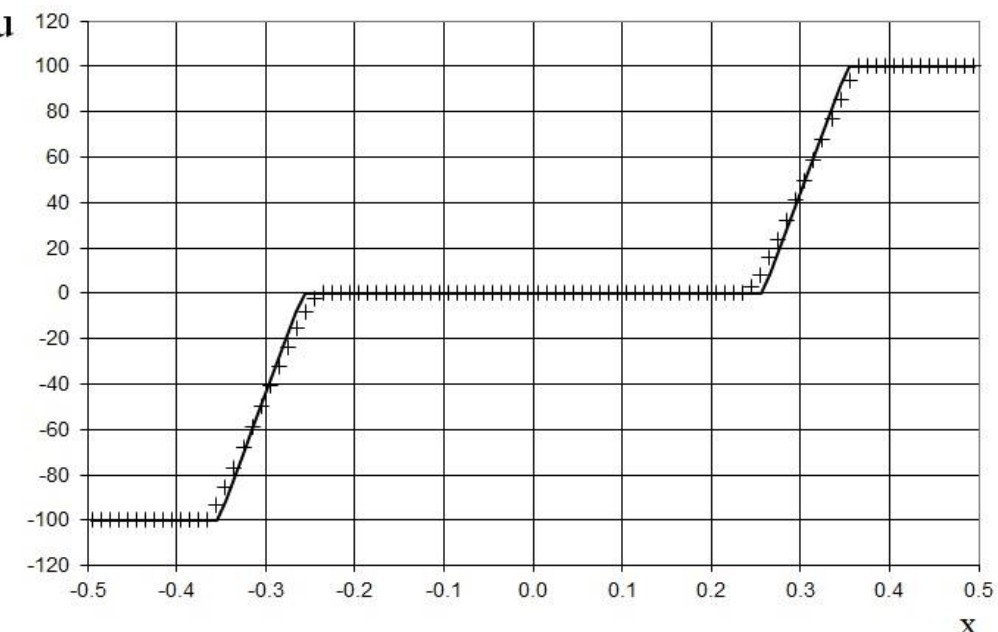

**Figure 18.** Velocity plot for the double expansion wave problem (47) for the one-dimensional scalar hyperbolic conservation law, using the coarse grid (100 grid points): the exact solution (line), and the present method (cross).

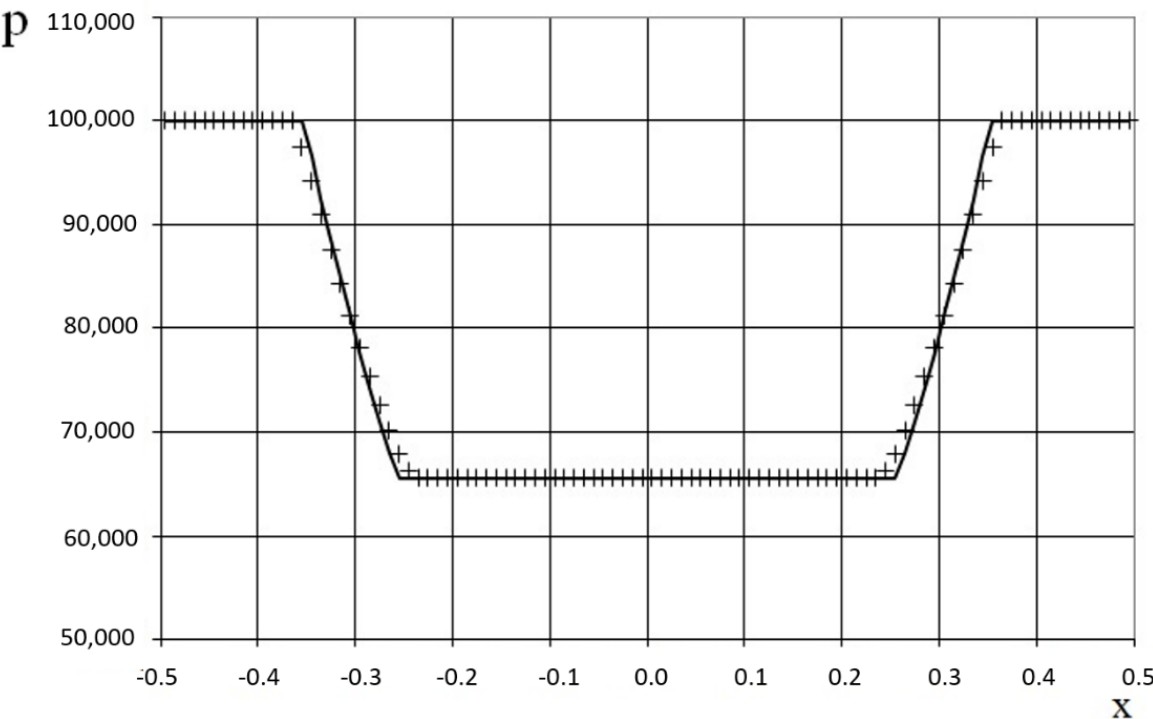

**Figure 19.** Pressure plot for the double expansion wave problem (47) for the one-dimensional scalar hyperbolic conservation law, using the coarse grid (100 grid points): the exact solution (line), and the present method (cross).

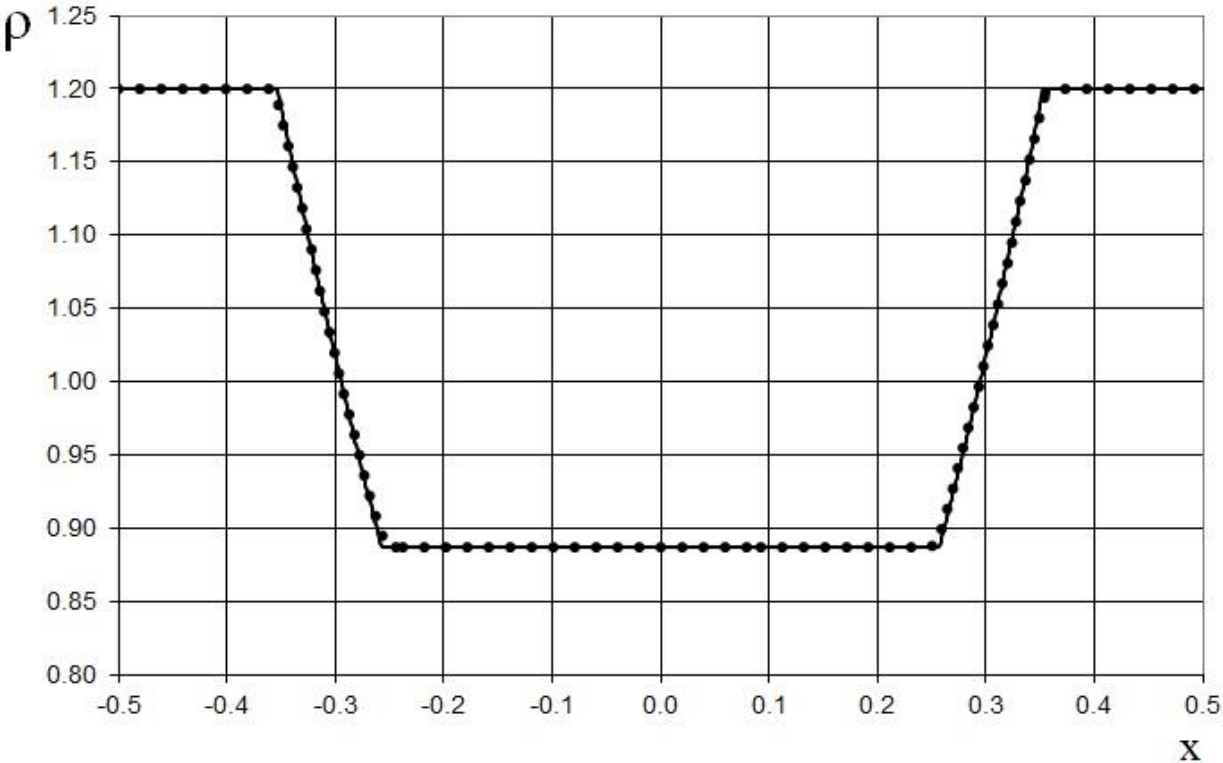

**Figure 20.** Density plot for the double expansion wave problem (47) for the one-dimensional scalar hyperbolic conservation law, using the finer grid (1000 grid points): the exact solution (solid line), and the present method (dotted line).

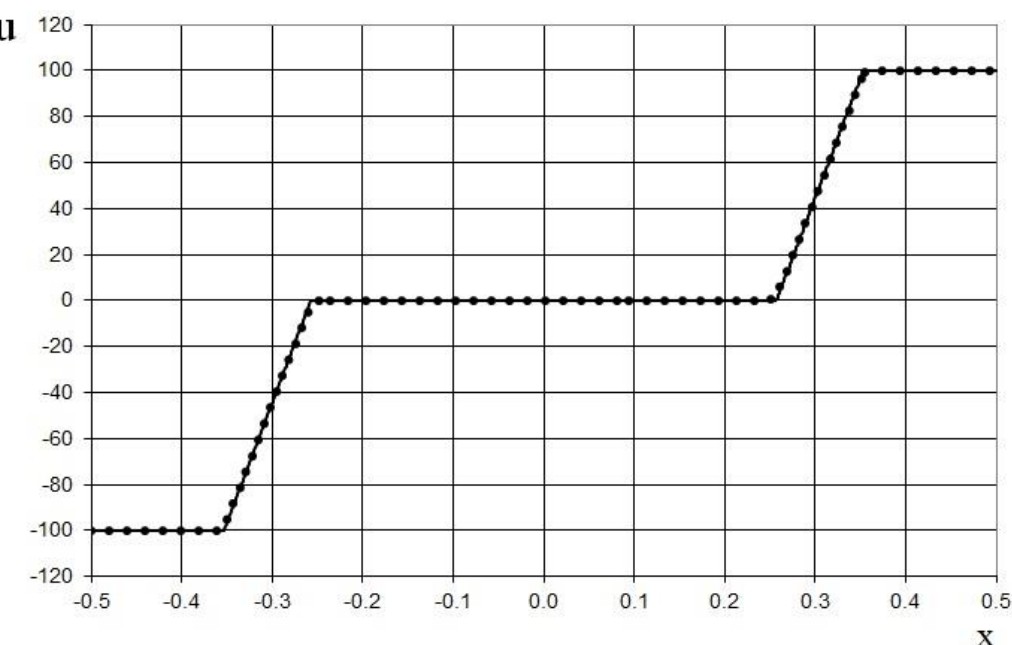

**Figure 21.** Velocity plot for the double expansion wave problem (47) for the one-dimensional scalar hyperbolic conservation law, using the finer grid (1000 grid points): the exact solution (solid line), and the present method (dotted line).

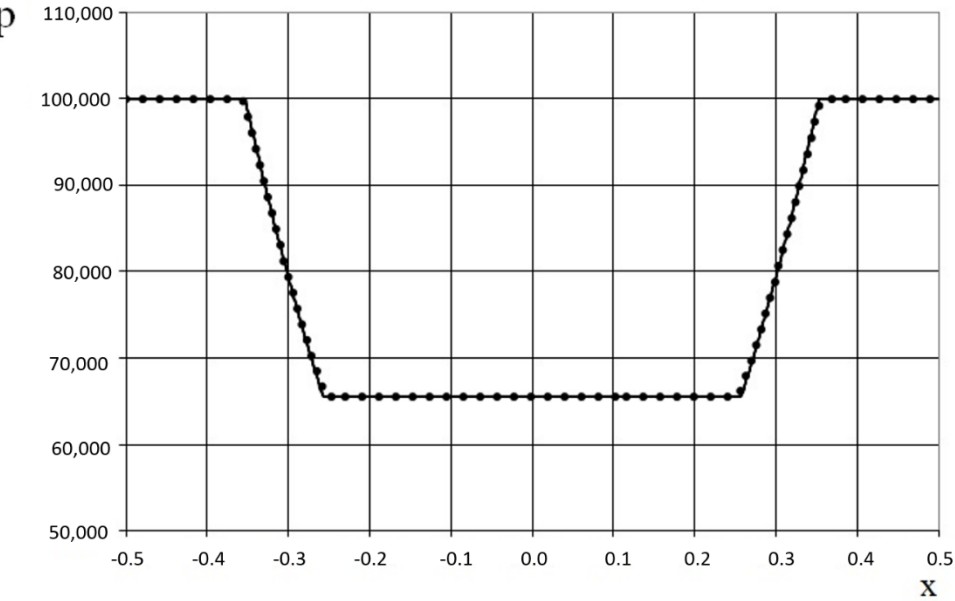

**Figure 22.** Pressure plot for the double expansion wave problem (47) for the one-dimensional scalar hyperbolic conservation law, using the finer grid (1000 grid points): the exact solution (solid line), and the present method (dotted line).

The fourth example is the Shu–Osher shock-tube problem.

The Shu–Osher problem [44] simulates a normal shock front moving inside a one-dimensional inviscid flow with artificial density fluctuations. The initial conditions for the simulation are

$$p_j^0 = 10.3333, \ u_j^0 = 2.629369, \ \rho_j^0 = 3.857143, \ \text{if } x_j < 1/8,$$
$$p_j^0 = 1.0, \ u_j^0 = 0, \ \rho_j^0 = 1 + 0.2 \sin(16\pi x), \ \text{if } x_j \geq 1/8.$$

(50)

The modeling area was taken to be $x \in [0, 1]$. Three kinds of grids were used: a coarse grid (192 grid points), a fine grid (384 grid points), and a finer grid (1000 grid points). The time step at the acoustics stage was chosen according to the Courant criterion (25), where $k_c = 2$—i.e., the shock propagates through two grid cells at the acoustics stage—for all grids. The time step at the convection stage was chosen according to (26), where $k_u = 5$—i.e., the shock propagates through five grid cells at the convection stage—for all grids. The marching time step was $_\Delta t = 0.010270978683$ for the coarse grid, $_\Delta t = 0.005135489341$ for the fine grid, and $_\Delta t = 0.001972027907$ for the finer grid. The acoustics and convection stages were performed at each marching time step, i.e., $_\Delta t_c = {_\Delta t_u} = {_\Delta t}$. To avoid additional fluctuations in the solution, we calculated the rarefaction wave at the acoustics stage instead of (A50) as

$$U_{R_j}^k = U_{R_i}^{k-1} + \frac{U_{R_{i+1}}^{k-1} - U_{R_i}^{k-1}}{x_2 - x_1}(x_j - x_1),$$

$$E_{R_j}^k = E_{R_i}^{k-1} + \frac{E_{R_{i+1}}^{k-1} - E_{R_i}^{k-1}}{x_2 - x_1}(x_j - x_1),$$

$$R_{R_j}^k = R_{R_i}^{k-1} + \frac{R_{R_{i+1}}^{k-1} - R_{R_i}^{k-1}}{x_2 - x_1}(x_j - x_1),$$

$$P_{R_j}^k = (\kappa - 1)R_{R_j}^k E_{R_j}^k,$$

(51)

and at the convection stage instead of (A67) as

$$u_j^k = u_i^{k-1} + \frac{u_{i+1}^{k-1} - u_i^{k-1}}{x_2 - x_1}(x_j - x_1),$$

$$\varepsilon_j^k = \varepsilon_i^{k-1} + \frac{\varepsilon_{i+1}^{k-1} - \varepsilon_i^{k-1}}{x_2 - x_1}(x_j - x_1),$$

$$\rho_j^k = \rho_i^{k-1} + \frac{\rho_{i+1}^{k-1} - \rho_i^{k-1}}{x_2 - x_1}(x_j - x_1),$$

$$p_j^k = (\kappa - 1)\rho_j^k \varepsilon_j^k.$$

(52)

The results are shown in Figures 23–25 in comparison with the "exact" solution [44] at the time $t = 0.178$. The results show that the fine grid and the finer grid are better than the coarse grid. However, owing to the accumulation of the roundoff error, the shock is delayed by a few grid cells for the coarse grid results at the shown moment in time.

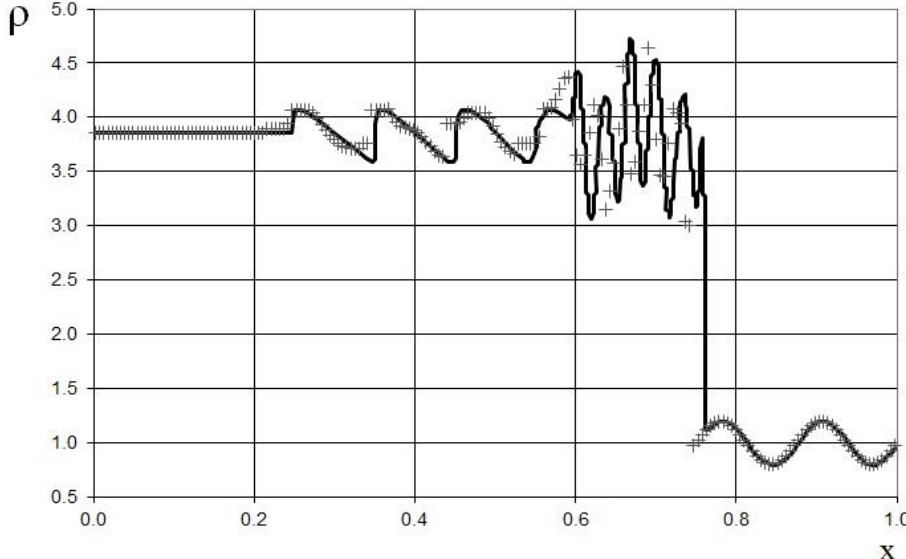

**Figure 23.** Density plot for the Shu–Osher shock-tube problem, using the coarse grid (192 grid points): the "exact" solution [44] (line), and the present method (cross).

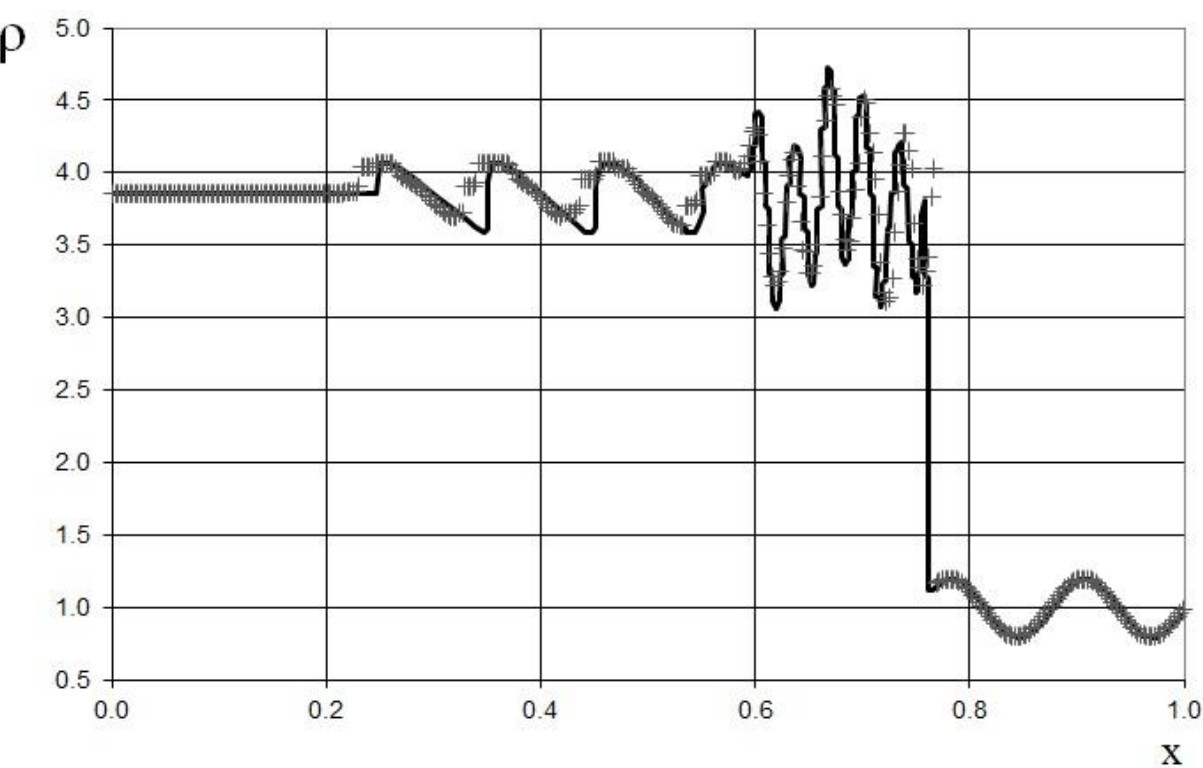

**Figure 24.** Density plot for the Shu–Osher shock-tube problem, using the fine grid (384 grid points): the "exact" solution [44] (line), and the present method (cross).

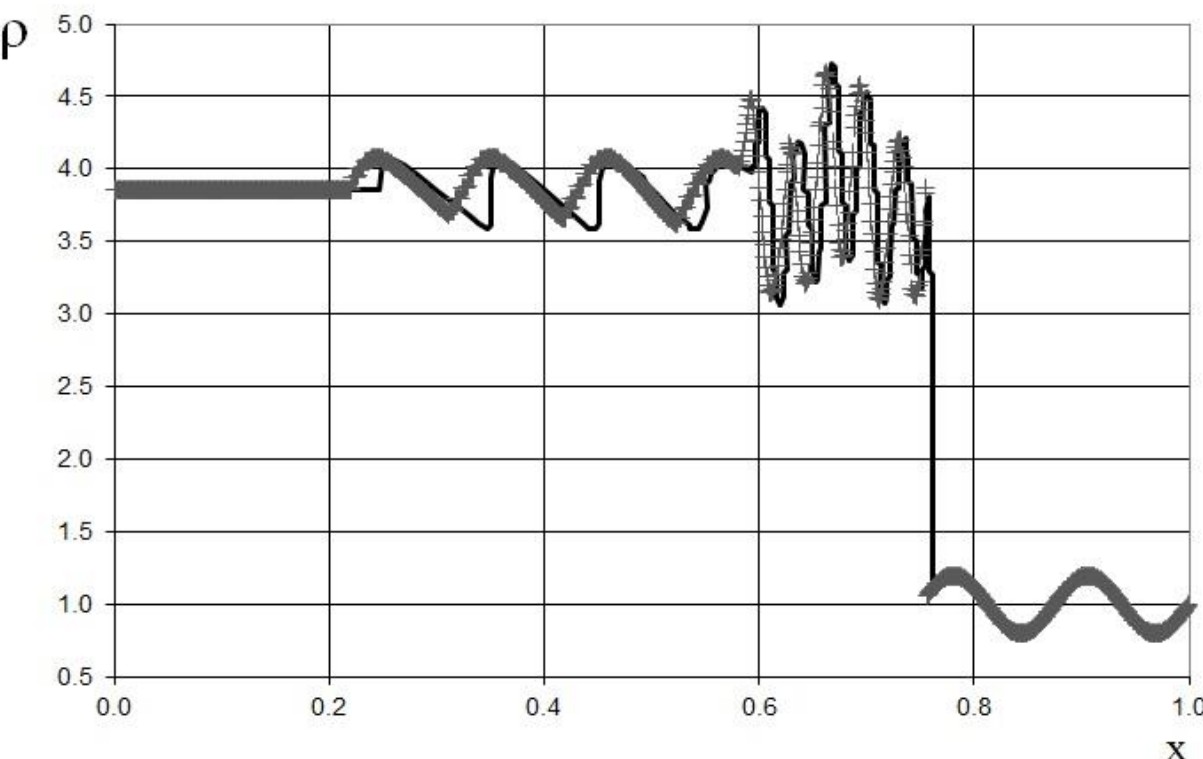

**Figure 25.** Density plot for the Shu–Osher shock-tube problem, using the finer grid (1000 grid points): the "exact" solution [44] (line), and the present method (cross).

The fifth example is a one-dimensional problem with initial conditions

$$p_j^0 = 101325.0, \; u_j^0 = 100.0, \; \rho_j^0 = 1.25. \tag{53}$$

In this case, at the time moment $t = 0$, a thin wall was lowered in the channel section $x = 0$.

The modeling area was taken to be $x \in [-5, 6]$. The grid contained 110 cells, and the grid step was $h = 0.1$. The time step at the acoustics stage was chosen according to the Courant criterion (25), where $k_c = 7$, i.e., the shock propagates through seven grid cells at the acoustics stage. The time step at the convection stage was chosen according to (26), where $k_u = 6$, i.e., the shock propagates through six grid cells at the convection stage. The marching time step was $_\Delta t = 0.002$. The acoustics stage was performed at each marching time step, i.e., $_\Delta t_c = {}_\Delta t$; the convection stage was performed after three steps, i.e., $_\Delta t_u = 3_\Delta t$.

The results are shown in Figures 26–28 in comparison with the exact solution at the time $t = 0.012$. Figures 26–28 show the absence of numerical viscosity for the shock. In this case, there is no delay caused by the shock.

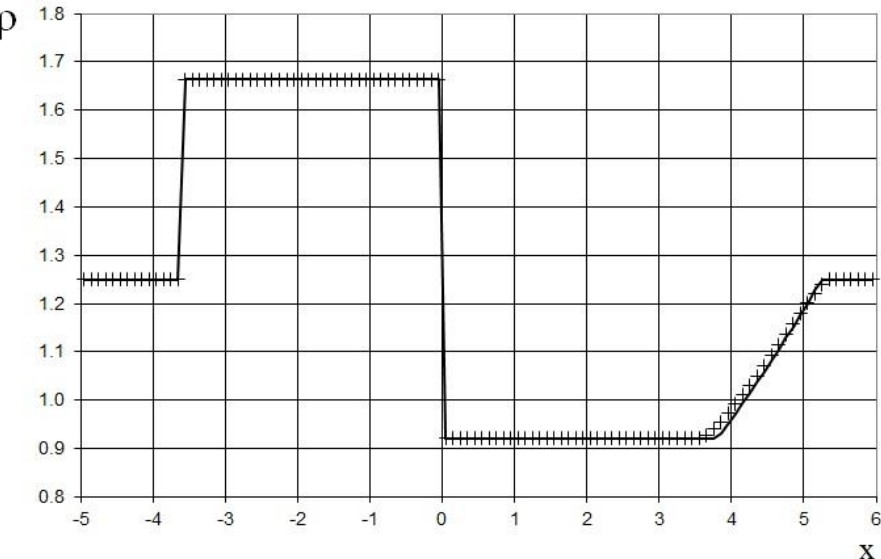

**Figure 26.** Density plot for the problem (53): the exact solution (line), and the present method (cross).

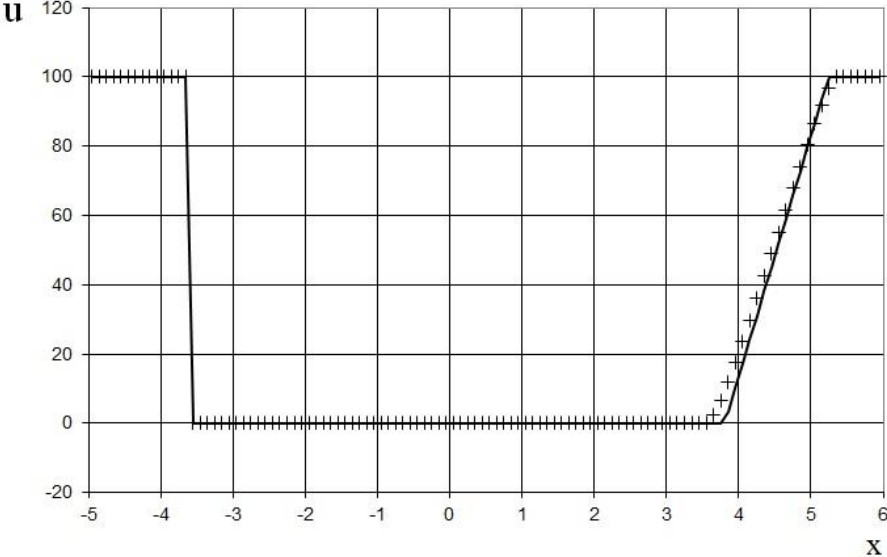

**Figure 27.** Velocity plot for the problem (53): the exact solution (line), and the present method (cross).

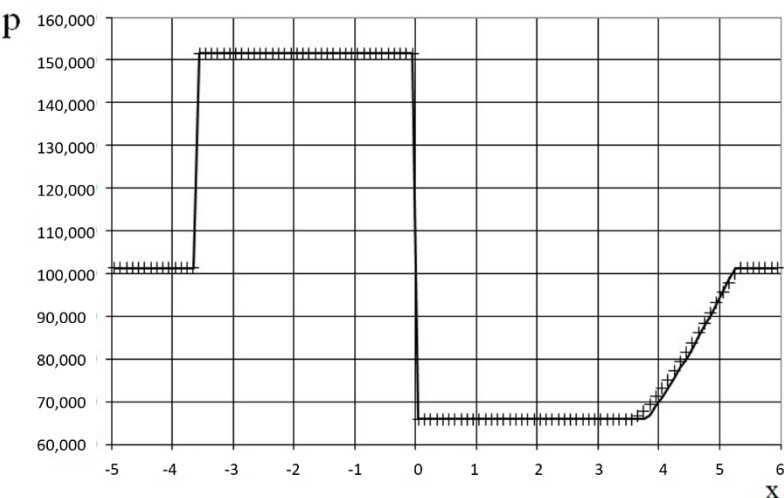

**Figure 28.** Pressure plot for the problem (53): the exact solution (line), and the present method (cross).

The sixth example is a one-dimensional problem with the initial conditions

$$p_j^0 = 101325.0, \quad u_j^0 = 600.0, \quad \rho_j^0 = 1.25. \tag{54}$$

In this case, as in the third problem, at the time $t = 0$, a thin wall was lowered in the channel section $x = 0$.

The modeling area was taken to be $x \in [-5, 5]$. The grid contained 100 cells, and the grid step was $h = 0.1$. The time step at the acoustics stage was chosen according to the Courant criterion (25), where $k_c = 2$, i.e., the shock propagates through two grid cells at the acoustics stage. The time step at the convection stage was chosen according to (26), where $k_u = 5$, i.e., the shock propagates through five grid cells at the convection stage. The marching time step was $\Delta t = 0.00079$. The acoustics and convection stages were performed at each marching time step, i.e., $\Delta t_c = \Delta t_u = \Delta t$.

The results are shown in Figures 29–31 in comparison with the exact solution at the time $t = 0.00474$. Figures 29–31 show the absence of numerical viscosity for the shock. In this case, there is no delay caused by the shock.

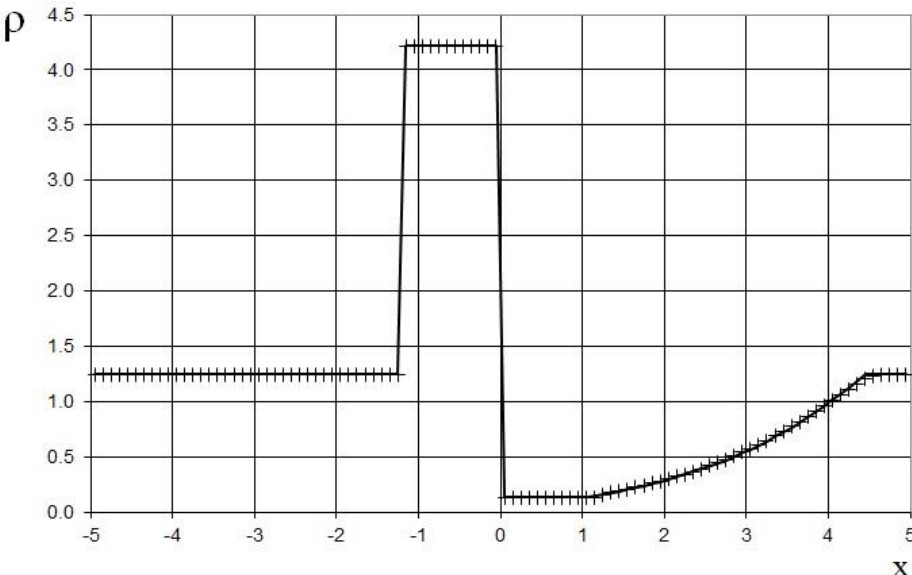

**Figure 29.** Density plot for the problem (54): the exact solution (line), and the present method (cross).

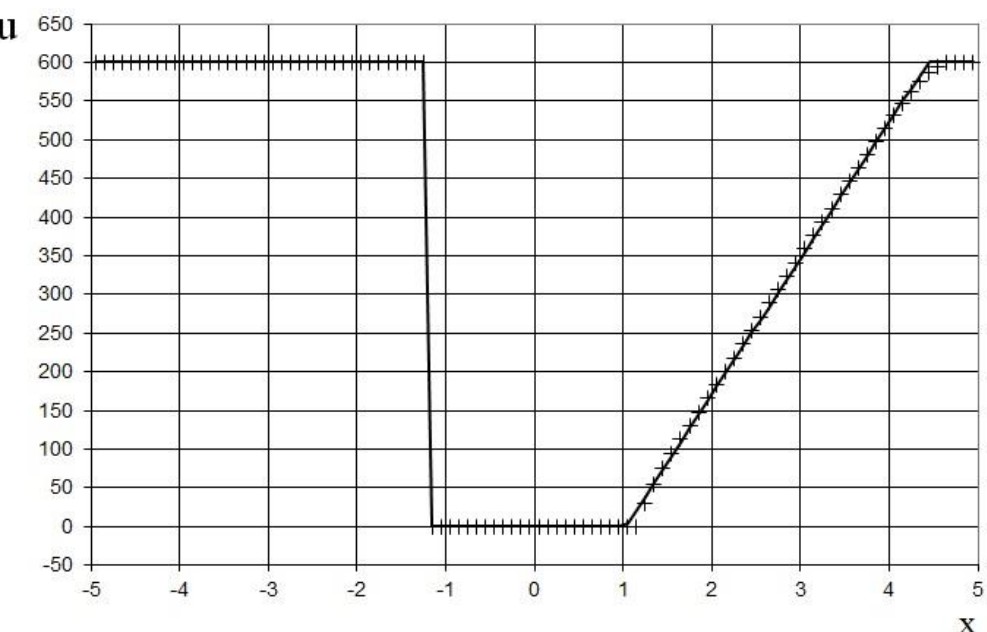

**Figure 30.** Velocity plot for the problem (54): the exact solution (line), and the present method (cross).

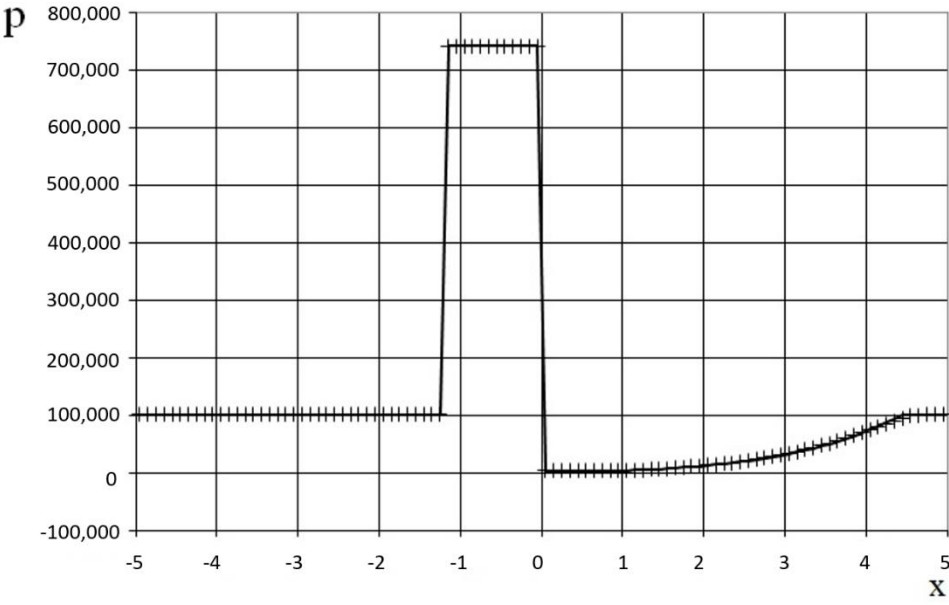

**Figure 31.** Pressure plot for the problem (54): the exact solution (line), and the present method (cross).

## 4. Discussion

The numerical solution results for the aforementioned well-known test cases were compared with the exact solutions and Harten's data. Based on the results obtained, the following conclusions can be drawn:

(1) The advantage of the proposed method in comparison with the known methods of TVD, ENO, and WENO is the absence of numerical viscosity (diffusion) for shocks;

(2) The convective terms are not neglected in comparison with the MOC methods, which is important in gas flow modeling;

(3) Over time, shocks or rarefaction waves can propagate somewhat slower or faster than in the exact solution. This can be attributed to the rounding off of the position of the wave fronts to the accuracy of the grid cell, owing to the use of the fixed homogeneous grid.

The iterative Godunov exact solver determines the accuracy of the proposed method for flow discontinuities. In calculations, we use the iteration termination condition less than $10^{-5}$ to find the pressure difference between the current and previous iterations.

The directions for further research are connected with two-dimensional modeling and the study of complex flow structures caused by fluid–solid interactions.

**Funding:** This research did not receive any specific grant from funding agencies in the public, commercial, or not-for-profit sectors.

**Acknowledgments:** I am grateful to V.G. Shakhov and other anonymous referees for their insightful comments and advice, which have greatly improved the quality of this work.

**Conflicts of Interest:** The author declares no conflict of interest.

## Nomenclature

| Symbol | Meaning |
| --- | --- |
| $u$ | Convective velocity of flow |
| $x$ | Direction |
| $t$ | Time |
| $\Delta t$ | Time step, marching time step |
| $e_1$ | Tolerance associated with the comparison of the flow variables |
| $e_2$ | Machine arithmetic tolerance associated with the comparison of the x coordinates |
| $h$ | Grid step |
| $n$ | Number of grid cells |
| $u_s$ | Convective velocity of a shock |
| $k_u$ | Integer coefficient in the formula for the time step at the convection stage |
| $a$ | Left boundary of a modeling area |
| $b$ | Right boundary of the modeling area |
| $\rho$ | Density |
| $p$ | Pressure |
| $\kappa$ | Ratio of specific heat coefficients |
| $c$ | Sound velocity |
| $w_i$ | Transfer variables of wave equations |
| $\overline{w}_i$ | Transfer vector variables |
| $i$ | Coordinate index |
| $j$ | Coordinate index |
| $k$ | Time index |
| $\widetilde{P}$ | Pressure function for adiabatic law |
| $\Delta t_c$ | time step at the acoustics stage |
| $c_s$ | Acoustic velocity of a shock |
| $k_c$ | Integer coefficient in the formula for the time step at the acoustics stage |
| $\Delta t_u$ | Time step at the convection stage |
| $n_c$ | An integer showing how many times the time step at the acoustics stage is greater than the marching time step |
| $n_u$ | An integer showing how many times the time step at the convection stage is greater than the marching time step |
| $\varepsilon$ | Internal specific energy |
| $[A]$ | Jacobian matrix |
| $[\Lambda]$ | Diagonal matrix of eigenvalues of the Jacobian matrix |
| $[L]$ | Matrix of corresponding left eigenvectors |
| $[R]$ | Matrix of corresponding right eigenvectors |
| $[E]$ | Diagonal identity matrix |
| $c*$ | Constant sound velocity |
| $\rho*$ | Constant density |
| $R_L$ | Density of the exact solution of the Riemann problem on the left |
| $R_R$ | Density of the exact solution of the Riemann problem on the right |
| $U_L$ | Velocity of the exact solution of the Riemann problem on the left |
| $U_R$ | Velocity of the exact solution of the Riemann problem on the right |

| | |
|---|---|
| $E_L$ | Internal specific energy of the Riemann problem on the left |
| $E_R$ | Internal specific energy of the Riemann problem on the right |
| $P_L$ | Pressure of the Riemann problem on the left |
| $P_R$ | Pressure of the Riemann problem on the right |
| $D_L$ | Velocity of the propagation of a shock or rarefaction wave of the Riemann problem on the left |
| $D_R$ | Velocity of the propagation of a shock or rarefaction wave of the Riemann problem on the right |
| $C_L$ | Local acoustic velocity of the Riemann problem on the left |
| $C_R$ | Local acoustic velocity of the Riemann problem on the right |
| $\rho_c$ | Density after the acoustics stage |
| $p_c$ | Pressure after the acoustics stage |
| $u_c$ | Velocity after the acoustics stage |
| $\varepsilon_c$ | Internal specific energy after the acoustics stage |
| $\rho_u$ | Density after the convection stage |
| $p_u$ | Pressure after the convection stage |
| $u_u$ | Velocity after the convection stage |
| $\varepsilon_u$ | Internal specific energy after the convection stage |
| $u_r$ | Velocity on the right |
| $\rho^0$ | Density for the initial conditions |
| $p^0$ | Pressure for the initial conditions |
| $u^0$ | Velocity for the initial conditions |
| $\pi$ | Pi |

## Appendix A. The Methodological Scheme for the Adiabatic Gas Dynamics Equations

*Appendix A.1. A Scheme of the Method for the Acoustics Stage*

At the acoustics stage, the system of Equation (23) is solved. These equations describe the movement of two waves with velocities $-c$ and $c$, and their respective transfer values, $w_1$ and $w_2$ (19). In the proposed method, we replace variables $w_1$ and $w_2$ with variables $\overline{w}_1 = \{w_1, u\}$ and $\overline{w}_2 = \{w_2, u\}$, which are transferred with local acoustic velocities $-c$ and $c$, respectively.

Consider a wave propagating along the positive direction of the *x*-axis (to the right). The grid cells are considered in pairs. The cells' coordinates after transferring at the acoustics stage for the right wave $\overline{w}_2$ are determined as follows:

$$
\begin{aligned}
x_1 &= x_i + c_i^{k-1} \Delta t_c, \\
x_2 &= x_{i+1} + c_{i+1}^{k-1} \Delta t_c.
\end{aligned}
\tag{A1}
$$

The solution to the problem is sought in the following form:

(1)  If conditions

$$
\left| w_2\,_{i+1}^{k-1} - w_2\,_i^{k-1} \right| < e_1, \quad \left| c_{i+1}^{k-1} - c_i^{k-1} \right| < e_1
\tag{A2}
$$

are satisfied, then for all grid cells for which the condition $x_1 \leq x_j \leq x_2$ is satisfied, the solution at the next time instant k is trivial, and can be given as follows:

$$
w_2\,_j^k = \frac{1}{2}(w_2\,_i^{k-1} + w_2\,_{i+1}^{k-1}), \quad u_j^k = \frac{1}{2}(u_i^{k-1} + u_{i+1}^{k-1}).
\tag{A3}
$$

(2)  If condition

$$
\left( c_{i+1}^{k-1} - c_i^{k-1} \right) < -e_1
\tag{A4}
$$

is satisfied, and
     (2.1) if

$$
(x_2 - x_1) > e_2,
\tag{A5}
$$

then we have a weak shock, which does not overtake the solution from the cell in front. Hence, the solution is given as follows:

$$w_{2\,j}^{\,k} = w_{2\,i}^{\,k-1},\ u_j^k = u_i^{k-1},\ \text{if } x_1 \le x_j \le (x_1 + x_2)/2,$$
$$w_{2\,j}^{\,k} = w_{2\,i+1}^{\,k-1},\ u_j^k = u_{i+1}^{k-1},\ \text{if } (x_1 + x_2)/2 < x_j \le x_2; \tag{A6}$$

(2.2) if

$$(x_2 - x_1) \le e_2, \tag{A7}$$

(2.2 a) and if

$$\left( u_{i+1}^{k-1} - u_i^{k-1} \right) \le 0 \text{ and } (x_1 - h/2) \le x_j \le (x_1 + h/2),$$

then the solution can be given as follows:

$$w_{2\,j}^{\,k} = w_{2\,i}^{\,k-1},\ u_j^k = u_i^{k-1}, \tag{A8}$$

because we have the shock, and
(2.2 b) if

$$\left( u_{i+1}^{k-1} - u_i^{k-1} \right) > 0 \text{ and } (x_2 - h/2) \le x_j \le (x_2 + h/2),$$

then

$$w_{2\,j}^{\,k} = w_{2\,i+1}^{\,k-1},\ u_j^k = u_{i+1}^{k-1}, \tag{A9}$$

because we have a rarefaction wave.

(3)    If

$$\left( c_{i+1}^{k-1} - c_i^{k-1} \right) \ge e_1,$$

and
(3.1) if

$$\left( u_{i+1}^{k-1} - u_i^{k-1} \right) < e_1,$$

then we have the shock.
(3.1 a) If

$$(x_1 - h/2) \le x_j \le (x_1 + h/2),$$

then

$$w_{2\,j}^{\,k} = w_{2\,i}^{\,k-1},\ u_j^k = u_i^{k-1}. \tag{A10}$$

(3.1 b) If

$$(x_1 + h/2) < x_j \le x_2,$$

then

$$w_{2\,j}^{\,k} = w_{2\,i+1}^{\,k-1},\ u_j^k = u_{i+1}^{k-1}. \tag{A11}$$

(3.2) If

$$\left( u_{i+1}^{k-1} - u_i^{k-1} \right) \ge e_1,$$

then we have a rarefaction wave, and the solution for $x_1 \le x_j \le x_2$ is determined as follows:

$$w_{2\,j}^{\,k} = w_{2\,i}^{\,k-1} + \frac{w_{2\,i+1}^{\,k-1} - w_{2\,i}^{\,k-1}}{x_2 - x_1}(x_j - x_1),$$
$$u_j^k = u_i^{k-1} + \frac{u_{i+1}^{k-1} - u_i^{k-1}}{x_2 - x_1}(x_j - x_1). \tag{A12}$$

Under conditions (1)–(3), if it turns out that the solution has already been assigned to the grid cell at the given substep, then it is replaced by a new one when the pressure of the new solution is greater than the pressure of the solution already in the grid cell. The pressure can be obtained by subtracting u from $w_2$; see (19). This is why for each Lagrangian cell we assigned not only $w_2$, but also u; thus, we replaced $w_2$ with $\overline{w}_2 = \{w_2, u\}$. Additionally, when assigning a solution to the grid cell, the condition is checked so that the propagation of these solutions with a local acoustic velocity does not overtake (or overwrite) the solution of the forward shock, if such a shock exists.

For the wave propagating along the negative direction of the *x*-axis (to the left), and with transfer values $\overline{w}_1 = \{w_1, u\}$, the conditions and expressions are obtained in a similar way.

After simulating the wave propagations along the left and right directions, we have sets of transfer values $\overline{w}_1 = \{w_1, u\}$ and $\overline{w}_2 = \{w_2, u\}$, respectively, in each cell of the grid. Hence, we can find the solution after the acoustic substep as follows:

$$u_i^k = 0.5(w_1{}_i^k - w_2{}_i^k),$$
$$\widetilde{P}_i^k = 0.5(w_2{}_i^k - w_1{}_i^k),$$

(A13)

where the density is determined from (20) as follows:

$$\rho_i^k = \rho_0 \left(1 + \widetilde{P}_i^k \frac{\kappa - 1}{2c_0}\right)^{\frac{2}{\kappa - 1}},$$

(A14)

and the pressure $p_i^k$ can be found from the adiabatic law (17). The solution to the acoustics stage is $\{p_c^k, u_c^k, \rho_c^k\}$.

*Appendix A.2. A Methodological Scheme for the Convection Stage*

At the convection stage, the system of Equation (24) is solved. We can replace this system with

$$\frac{\partial \overline{w}}{\partial t} + u \frac{\partial \overline{w}}{\partial x} = 0,$$

(A15)

where $\overline{w} = \{\rho, u\}$. The solution of the acoustics stage $\{\rho_c^k, u_c^k\}$ gives the initial conditions for the convection stage $\overline{w} = \{\rho_u^{k-1}, u_u^{k-1}\} = \{\rho_c^k, u_c^k\}$, which are transferred with local convective velocity $u_u^{k-1}$. Additionally, the cells are considered in pairs at the convection stage. The cells' coordinates after transferring at the convection stage are determined as follows:

$$x_1 = x_i + u_{ui}^{k-1} \Delta t_u,$$
$$x_2 = x_{i+1} + u_{ui+1}^{k-1} \Delta t_u.$$

(A16)

Furthermore, after we omit subscript u, the variables here refer to the convection stage. The solution at the convection stage is sought in the following form:

(1)   if the conditions

$$|u_{i+1} - u_i| < e_1, \ |\rho_{i+1} - \rho_i| < e_1$$

(A17)

are satisfied, then for all grid cells for which the condition is satisfied, the solution at the next time moment k is trivial:

$$\rho_j^k = \frac{1}{2}(\rho_i^{k-1} + \rho_{i+1}^{k-1}), \ u_j^k = \frac{1}{2}(u_i^{k-1} + u_{i+1}^{k-1}).$$

(A18)

When assigning the solution (A18) to the grid cells, the condition is separately checked so that the propagation of this solution at the convection velocity does not overtake (does not overwrite) the solution of the forward shock, if one exists. For this, at the beginning of the

convection stage, the positions of all of the shock boundaries at the next moment in time are determined.

(2)   If the conditions

$$(u_{i+1} - u_i) < e_1 \tag{A19}$$

are satisfied, then we have the shock.

(2.1) If

$$(x_2 - x_1) > e_2, \tag{A20}$$

then the solution is defined as follows:

(2.1 a) if

$$x_1 \leq x_j \leq (x_1 + x_2)/2,$$

then

$$\rho_j^k = \rho_i^{k-1}, \; u_j^k = u_i^{k-1}; \tag{A21}$$

(2.1 b) if

$$(x_1 + x_2)/2 < x_j \leq x_2,$$

then

$$\rho_j^k = \rho_{i+1}^{k-1}, \; u_j^k = u_{i+1}^{k-1}; \tag{A22}$$

(2.2) if

$$(x_2 - x_1) \leq e_2, \tag{A23}$$

then

(2.2 a) if

$$\left| p_{i+1}^{k-1} - p_i^{k-1} \right| < e_1,$$

and the discontinuity boundary is first determined as follows:

$$x = x_i + h/2 + (u_i^{k-1} + u_{i+1}^{k-1}) \, _\Delta t_u, \tag{A24}$$

then the solution is assigned to cell j to the left of the discontinuity boundary

$$\rho_j^k = \rho_i^{k-1}, \; u_j^k = u_i^{k-1}, \tag{A25}$$

and in cell j + 1 to the right of the discontinuity boundary, the values are assigned as follows:

$$\rho_j^k = \rho_{i+1}^{k-1}, \; u_j^k = u_{i+1}^{k-1}. \tag{A26}$$

(2.2 b) If

$$\left| p_{i+1}^{k-1} - p_i^{k-1} \right| \geq e_1,$$

then the solution is defined as follows:

-   if

$$p_i^{k-1} > p_{i+1}^{k-1} \text{ and } (x_1 - h/2) \leq x_j \leq (x_1 + h/2),$$

then

$$\rho_j^k = \rho_i^{k-1}, \; u_j^k = u_i^{k-1}; \tag{A27}$$

-   if

$$p_i^{k-1} < p_{i+1}^{k-1} \text{ and } (x_2 - h/2) \leq x_j \leq (x_2 + h/2),$$

then

$$\rho_j^k = \rho_{i+1}^{k-1}, \; u_j^k = u_{i+1}^{k-1}. \tag{A28}$$

(3)    If

$$(u_{i+1} - u_i) \geq e_1, \tag{A29}$$

then we have the rarefaction wave, and the solution for $x_1 \leq x_j \leq x_2$ is determined from the conservation condition of the Riemann invariants as follows:

$$u_j^k = u_i^{k-1} + \frac{u_{i+1}^{k-1} - u_i^{k-1}}{x_2 - x_1}(x_j - x_1),$$

$$\rho_j^k = \left( \left( \rho_i^{k-1} \right)^{\frac{\kappa-1}{2}} + \frac{\left( \rho_{i+1}^{k-1} \right)^{\frac{\kappa-1}{2}} - \left( \rho_i^{k-1} \right)^{\frac{\kappa-1}{2}}}{x_2 - x_1}(x_j - x_1) \right)^{\frac{2}{\kappa-1}}. \tag{A30}$$

When conditions (1)–(3) are satisfied, if it turns out that the solution has already been assigned to the grid cell at this substep, then it is replaced by a new one when the pressure of the new solution is greater than the pressure of the solution already in the grid cell.

## Appendix B. The Scheme of the Method for the One-Dimensional Scalar Hyperbolic Conservation Law

*Appendix B.1. A Methodological Scheme for the Acoustics Stage of the One-Dimensional Scalar Hyperbolic Conservation Law*

At the acoustics stage, the system of Equation (30) is solved. Although we cannot obtain the system of wave Equation (39) for nonlinear cases, the fundamental properties of the solution of the linear system (39) are conserved for the solution of the nonlinear system (30). The solutions for the nonlinear system are a shock wave or a rarefaction wave [37], just like for the linear system. Unfortunately, we do not know the variables $w_1$ and $w_3$ for the nonlinear system (30), but we know the exact solution of the Riemann problem using the Godunov method [25] for the nonlinear system (28), (29). We denote this solution with the large variables: $R_L$ and $R_R$—density, $U_L = U_R$—velocity, $E_L$ and $E_R$—internal specific energy, $P_L = P_R$—pressure, and $D_L$ and $D_R$—velocity of propagation of a shock or rarefaction wave, where the subscripts L and R denote the left and right regions of an edge between the cells, respectively.

In the proposed method, we replace the variables $w_1$ and $w_3$ (40) with the variables $\overline{w}_1 = \{P_L, U_L, R_L, E_L\}$ and $\overline{w}_3 = \{P_R, U_R, R_R, E_R\}$, which are transferred with local acoustic velocities $C_L = D_L - U_L$ and $C_R = D_R - U_R$, respectively.

We first need to solve the Riemann problem using the Godunov method [25] for every pair of cells (i and i + 1) with data ($\{p_i, u_i, \rho_i, \varepsilon_i\}$ and $\{p_{i+1}, u_{i+1}, \rho_{i+1}, \varepsilon_{i+1}\}$), respectively. As a result, the values of the large variables $\{P, U, R_L, E_L, D_L, R_R, E_R, D_R\}$ are obtained, which are assigned to the grid cells as follows:

$$P_{L_{i+1}}^{k-1} = P, \ U_{L_{i+1}}^{k-1} = U, \ R_{L_{i+1}}^{k-1} = R_L, \ E_{L_{i+1}}^{k-1} = E_L, \ C_{L_{i+1}}^{k-1} = D_L - U, \tag{A31}$$

which constitute the vector $\overline{w}_1$ of the wave propagating along the negative direction of the *x*-axis (to the left); and:

$$P_{R_i}^{k-1} = P, \ U_{R_i}^{k-1} = U, \ R_{R_i}^{k-1} = R_R, \ E_{R_i}^{k-1} = E_R, \ C_{R_i}^{k-1} = D_R - U, \tag{A32}$$

which constitute the vector $\overline{w}_3$ of the wave propagating along the positive direction of the *x*-axis (to the right). In the case of a rarefaction wave, the variable D assigns the velocity of the slower characteristic of the rarefaction wave. For example, for the right rarefaction wave, $D_R$ is given as follows:

$$D_R = U + c_{i+1} - \frac{(\kappa - 1)}{2}(u_{i+1} - U), \tag{A33}$$

where

$$c_{i+1} = \sqrt{\kappa \frac{P_{i+1}}{\rho_{i+1}}}, \tag{A34}$$

and for the left rarefaction wave $D_L$ is determined as

$$D_L = U - c_i - \frac{(\kappa - 1)}{2}(u_i - U).D_L = U - c_i - \frac{(\kappa - 1)}{2}(u_i - U). \tag{A35}$$

Consider a wave propagating along the positive direction of the *x*-axis (to the right). The grid cells are considered in pairs. The cells' coordinates after transferring at the acoustics stage for the right wave $\overline{w}_3$ are determined as follows:

$$x_1 = x_i + C_{R_i}^{k-1} \Delta t_c, \tag{A36}$$
$$x_2 = x_{i+1} + C_{R_{i+1}}^{k-1} \Delta t_c$$

The solution to the problem is sought in the following form:

(1) If the conditions

$$\left| P_{R_{i+1}}^{k-1} - P_{R_i}^{k-1} \right| < e_1, \ \left| C_{R_{i+1}}^{k-1} - C_{R_i}^{k-1} \right| < e_1 \tag{A37}$$

are satisfied, then for all grid cells for which the condition $x_1 \leq x_j \leq x_2$ is satisfied, the solution at the next time instant k is trivial, and can be given as follows:

$$P_{R_j}^k = \tfrac{1}{2}(P_{R_i}^{k-1} + P_{R_{i+1}}^{k-1}), \ U_{R_j}^k = \tfrac{1}{2}(U_{R_i}^{k-1} + U_{R_{i+1}}^{k-1}),$$
$$R_{R_j}^k = \tfrac{1}{2}(R_{R_i}^{k-1} + R_{R_{i+1}}^{k-1}), \ E_{R_j}^k = \tfrac{1}{2}(E_{R_i}^{k-1} + E_{R_{i+1}}^{k-1}). \tag{A38}$$

(2) If the condition

$$\left( C_{R_{i+1}}^{k-1} - C_{R_i}^{k-1} \right) < -e_1 \tag{A39}$$

is satisfied, and
   (2.1) if

$$(x_2 - x_1) > e_2, \tag{A40}$$

then we have a weak shock, which does not overtake the solution from the cell in front. Hence, the solution is given as follows:

$$P_{R_j}^k = P_{R_i}^{k-1}, \ U_{R_j}^k = U_{R_i}^{k-1}, \ R_{R_j}^k = R_{R_i}^{k-1}, \ E_{R_j}^k = E_{R_i}^{k-1}, \text{ if } x_1 \leq x_j \leq (x_1 + x_2)/2,$$
$$P_{R_j}^k = P_{R_{i+1}}^{k-1}, \ U_{R_j}^k = U_{R_{i+1}}^{k-1}, \ R_{R_j}^k = R_{R_{i+1}}^{k-1}, \ E_{R_j}^k = E_{R_{i+1}}^{k-1}, \text{ if } (x_1 + x_2)/2 < x_j \leq x_2; \tag{A41}$$

   (2.2) if

$$(x_2 - x_1) \leq e_2, \tag{A42}$$

   (2.2 a) if

$$\left( U_{R_{i+1}}^{k-1} - U_{R_i}^{k-1} \right) \leq 0 \text{ and } (x_1 - h/2) \leq x_j \leq (x_1 + h/2),$$

then the solution can be given as follows:

$$P_{R_j}^k = P_{R_i}^{k-1}, \ U_{R_j}^k = U_{R_i}^{k-1}, \ R_{R_j}^k = R_{R_i}^{k-1}, \ E_{R_j}^k = E_{R_i}^{k-1}, \tag{A43}$$

because we have the shock, and

(2.2 b) if

$$\left(U_{R_{i+1}}^{k-1} - U_{R_i}^{k-1}\right) > 0 \text{ and } (x_2 - h/2) \le x_j \le (x_2 + h/2),$$

then

$$P_{R_j}^k = P_{R_{i+1}}^{k-1}, \ U_{R_j}^k = U_{R_{i+1}}^{k-1}, \ R_{R_j}^k = R_{R_{i+1}}^{k-1}, \ E_{R_j}^k = E_{R_{i+1}}^{k-1} \tag{A44}$$

because we have a rarefaction wave.

(3)    If

$$\left(C_{R_{i+1}}^{k-1} - C_{R_i}^{k-1}\right) \ge e_1, \tag{A45}$$

then

(3.1) if

$$\left(U_{R_{i+1}}^{k-1} - U_{R_i}^{k-1}\right) < e_1,$$

then we have the shock.

(3.1 a) If

$$(x_1 - h/2) \le x_j \le (x_1 + h/2),$$

then

$$P_{R_j}^k = P_{R_i}^{k-1}, \ U_{R_j}^k = U_{R_i}^{k-1}, \ R_{R_j}^k = R_{R_i}^{k-1}, \ E_{R_j}^k = E_{R_i}^{k-1}; \tag{A46}$$

(3.1 b) if

$$(x_1 + h/2) < x_j \le x_2,$$

then

$$P_{R_j}^k = P_{R_{i+1}}^{k-1}, \ U_{R_j}^k = U_{R_{i+1}}^{k-1}, \ R_{R_j}^k = R_{R_{i+1}}^{k-1}, \ E_{R_j}^k = E_{R_{i+1}}^{k-1}. \tag{A47}$$

This option is possible because of the following ratio of the local acoustic velocity of the shock to the sound velocity of the unperturbed flow ahead of its front, which can be given as follows:

$$\frac{C_{R_i}^{k-1}}{C_{R_{i+1}}^{k-1}} = \frac{D_R - U}{c_R} = \frac{\rho_R}{R_R}\sqrt{\frac{(\kappa+1)P}{2\kappa \, p_R} + \frac{\kappa-1}{2\kappa}} \tag{A48}$$

or

$$\frac{D_R - U}{c_R} = \frac{\rho_R}{R_R}\sqrt{\frac{(\kappa+1)}{2\kappa}\left(\frac{\kappa+1-(\kappa-1)\frac{\rho_R}{R_R}}{(\kappa+1)\frac{\rho_R}{R_R}-(\kappa-1)}\right) + \frac{\kappa-1}{2\kappa}}. \tag{A49}$$

From the graphical analysis of this ratio, it follows that for weak shock waves, it will be less than unity and increase indefinitely.

Particularly, the test case of a shock wave considered by Sod [42] refers to the variant "(3.1)."

(3.2) If

$$\left(U_{R_{i+1}}^{k-1} - U_{R_i}^{k-1}\right) \ge e_1,$$

then we have a rarefaction wave, and the solution for $x_1 \leq x_j \leq x_2$ is determined from the Riemann invariants of the hyperbolic conservation law as follows:

$$U_{R_j}^k = U_{R_i}^{k-1} + \frac{U_{R_{i+1}}^{k-1} - U_{R_i}^{k-1}}{x_2 - x_1}(x_j - x_1),$$

$$E_{R_j}^k = \left( \sqrt{E_{R_i}^{k-1}} + \frac{\sqrt{E_{R_{i+1}}^{k-1}} - \sqrt{E_{R_i}^{k-1}}}{x_2 - x_1}(x_j - x_1) \right)^2,$$

$$R_{R_j}^k = \left( R_{R_i}^{k-1}\left(\frac{E_{R_j}^k}{E_{R_i}^{k-1}}\right)^{\frac{1}{\kappa-1}} + R_{R_{i+1}}^{k-1}\left(\frac{E_{R_j}^k}{E_{R_{i+1}}^{k-1}}\right)^{\frac{1}{\kappa-1}} \right)/2,$$

$$P_{R_j}^k = (\kappa - 1)R_{R_j}^k E_{R_j}^k.$$

(A50)

Under conditions (1)–(3), if it turns out that the solution has already been assigned to the grid cell at the given substep, then it is replaced by a new solution obtained using the Riemann exact solver. In this case, the solution already assigned will be with the subscript R, and the next solution that comes to this cell will be with the subscript L. The resulting new solution is taken from the result of the Riemann problem with subscript R. Additionally, when assigning a solution to the grid cell, the condition is checked so that the propagation of these solutions with a local acoustic velocity does not overtake (or overwrite) the solution of the forward shock, if such a shock exists.

For the wave propagating along the negative direction of the *x*-axis (to the left), and with transfer values $\overline{w}_1 = \{P_L, U_L, R_L, E_L\}$, the conditions and expressions are obtained in a similar way.

After simulating the wave propagations along the left and right directions, we have sets of transfer values $\overline{w}_1 = \{P_L^k, U_L^k, R_L^k, E_L^k\}$ and $\overline{w}_3 = \{P_R^k, U_R^k, R_R^k, E_R^k\}$, respectively, in each cell of the grid, as well as a solution from the last substep in time $\{p^{k-1}, u^{k-1}, \rho^{k-1}, \varepsilon^{k-1}\}$ (see Figure A1) for which the Riemann problem is solved by the Godunov exact solver [25] three times. The first time the problem is solved for data $\overline{w}_3 = \{P_R^k, U_R^k, R_R^k, E_R^k\}$ on the left and $\{p^{k-1}, u^{k-1}, \rho^{k-1}, \varepsilon^{k-1}\}$ on the right (see Figure A2), the resulting right solution is assigned to the data of the Riemann problem solved for the third time from the left (see Figure A2). The second time the problem is solved for the data $\{p^{k-1}, u^{k-1}, \rho^{k-1}, \varepsilon^{k-1}\}$ on the left and $\overline{w}_1 = \{P_L^k, U_L^k, R_L^k, E_L^k\}$ on the right (see Figure A3), the resulting left solution is assigned to the data of the third problem on the right (see Figure A4). Finally, the third time the problem is solved for the aforementioned data (see Figure A4), as a result, we again obtain the values of the large variables $\{P^{k*}, U^{k*}, R_L^{k*}, E_L^{k*}, R_R^{k*}, E_R^{k*}\}$.

Because the pressure and convective velocity are equal for the left and right sides, their values after the acoustics stage are determined as follows:

$$p_{ci}^k = p_i^{k*}, \quad u_{ci}^k = U_i^{k*}.$$

(A51)

The values for density and energy are selected as follows:

$$\rho_{ci}^k = R_{R_i}^{k*}, \quad \varepsilon_{ci}^k = E_{R_i}^{k*}, \text{ if } U_i^{k*} \geq 0,$$

$$\rho_{ci}^k = R_{L_i}^{k*}, \quad \varepsilon_{ci}^k = E_{L_i}^{k*}, \text{ if } U_i^{k*} < 0.$$

(A52)

The solution to the acoustics stage is $\{p_c^k, u_c^k, \rho_c^k, \varepsilon_c^k\}$.

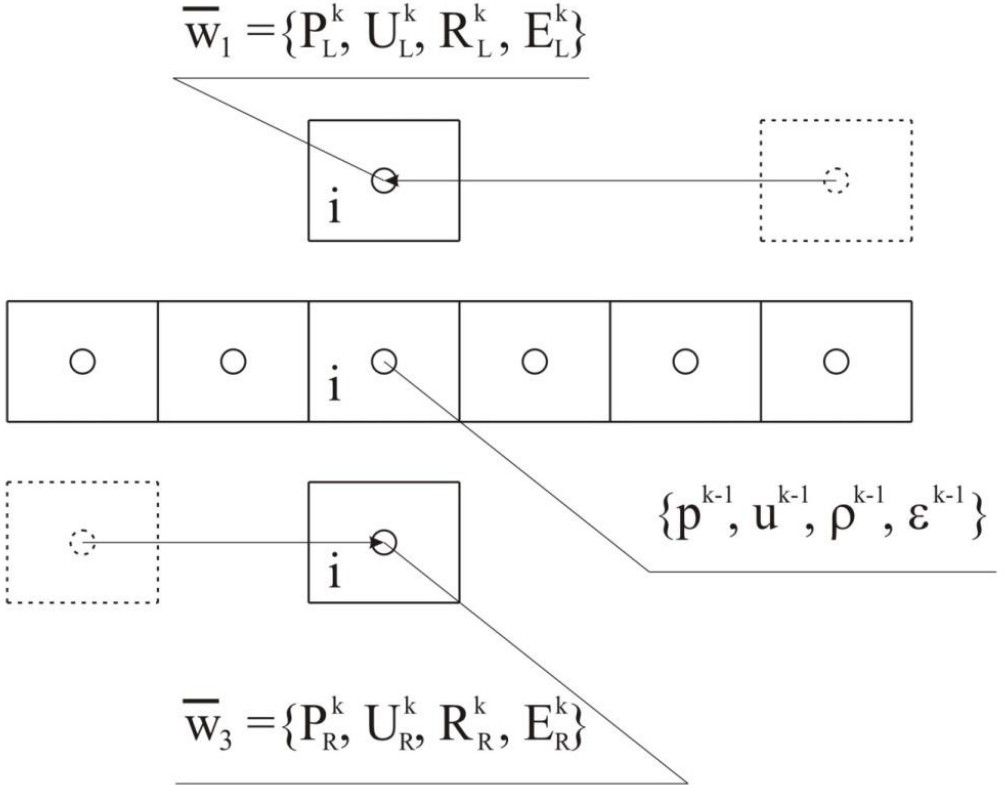

**Figure A1.** Initial conditions for the three Riemann problems.

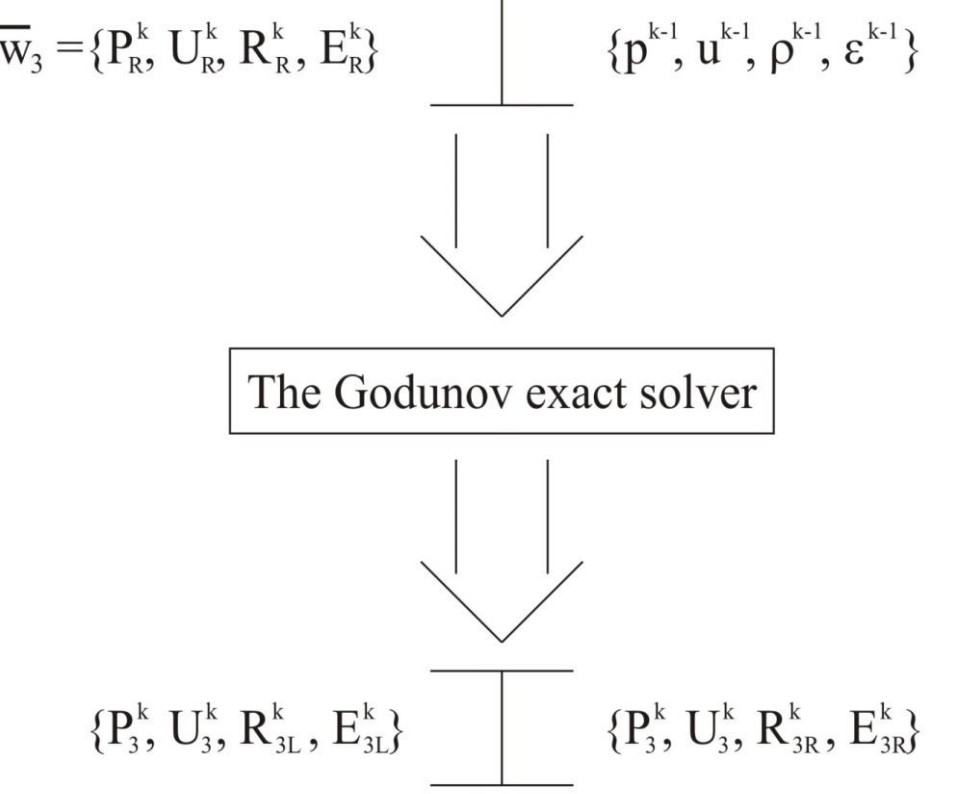

**Figure A2.** The first Riemann problem.

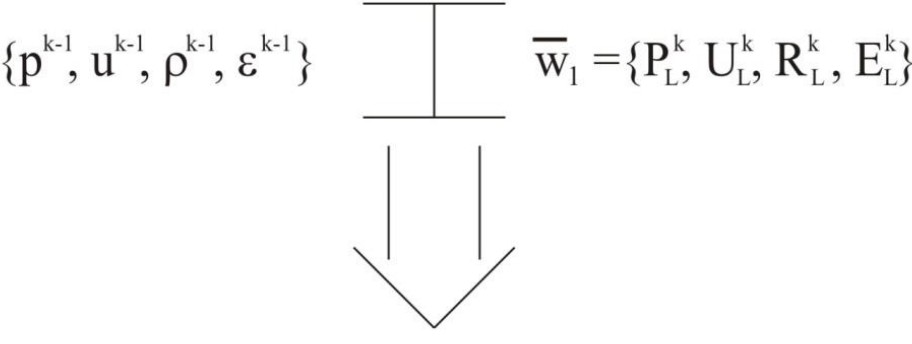

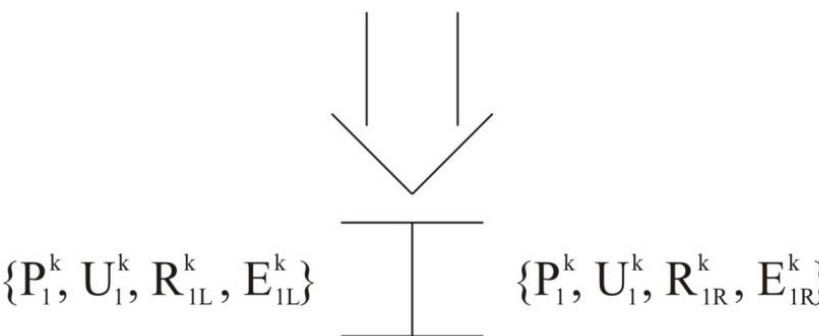

**Figure A3.** The second Riemann problem.

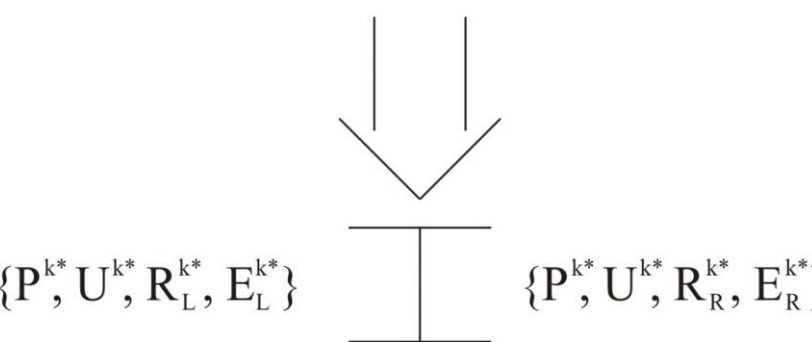

**Figure A4.** The third Riemann problem.

*Appendix B.2. A Scheme of the Method for the Convection Stage of the One-Dimensional Scalar Hyperbolic Conservation Law*

At the convection stage, the system of Equations (29) and (41) is solved. In the proposed method, the value $\{p_c^k, u_c^k, \rho_c^k, \varepsilon_c^k\} = \{p_u^{k-1}, u_u^{k-1}, \rho_u^{k-1}, \varepsilon_u^{k-1}\}$ is transferred with local convective velocity $u_c^k$, i.e., the calculated results of the acoustics stage are the initial data for the convection stage. Additionally, the cells are considered in pairs at the convection stage. The cells' coordinates after transferring at the convection stage are determined as follows:

$$x_1 = x + u_{ui}^{k-1} \Delta t_u,$$
$$x_2 = x_{i+1} + u_{ui+1}^{k-1} \Delta t_u. \tag{A53}$$

Furthermore, after we omit subscript u, the variables here refer to the convection stage. The solution at the convection stage is sought in the following form:

(1)　if the conditions

$$|u_{i+1} - u_i| < e_1, \; |\rho_{i+1} - \rho_i| < e_1, \; |\varepsilon_{i+1} - \varepsilon_i| < e_1 \tag{A54}$$

are satisfied, then for all grid cells for which the condition is satisfied, the solution at the next time moment k is trivial:

$$p_j^k = \frac{1}{2}(p_i^{k-1} + p_{i+1}^{k-1}), \; u_j^k = \frac{1}{2}(u_i^{k-1} + u_{i+1}^{k-1}), \; \rho_j^k = \frac{1}{2}(\rho_i^{k-1} + \rho_{i+1}^{k-1}), \; \varepsilon_j^k = \frac{1}{2}(\varepsilon_i^{k-1} + \varepsilon_{i+1}^{k-1}). \tag{A55}$$

When assigning the solution (A55) to the grid cells, the condition is separately checked so that the propagation of this solution at the convection velocity does not overtake (does not overwrite) the solution of the forward shock, if one exists. For this, at the beginning of the convection stage, the positions of all of the shock boundaries at the next moment in time are determined.

(2)　If the conditions

$$(u_{i+1} - u_i) < e_1 \tag{A56}$$

are satisfied, then we have the shock.

(2.1) If

$$(x_2 - x_1) > e_2, \tag{A57}$$

then the solution is defined as follows:

(2.1 a) if

$$x_1 \le x_j \le (x_1 + x_2)/2,$$

then

$$p_j^k = p_i^{k-1}, \; u_j^k = u_i^{k-1}, \; \rho_j^k = \rho_i^{k-1}, \; \varepsilon_j^k = \varepsilon_i^{k-1}; \tag{A58}$$

(2.1 b) if

$$(x_1 + x_2)/2 < x_j \le x_2,$$

then

$$p_j^k = p_{i+1}^{k-1}, \; u_j^k = u_{i+1}^{k-1}, \; \rho_j^k = \rho_{i+1}^{k-1}, \; \varepsilon_j^k = \varepsilon_{i+1}^{k-1}; \tag{A59}$$

(2.2) if

$$(x_2 - x_1) \le e_2, \tag{A60}$$

then

(2.2 a) if

$$\left| p_{i+1}^{k-1} - p_i^{k-1} \right| < e_1,$$

then the discontinuity boundary is first determined as follows:

$$x = x_i + h/2 + (u_i^{k-1} + u_{i+1}^{k-1})\,_\Delta t_u \tag{A61}$$

and the solution is assigned to cell j to the left of the discontinuity boundary

$$p_j^k = p_i^{k-1}, \ u_j^k = u_i^{k-1}, \ \rho_j^k = \rho_i^{k-1}, \ \varepsilon_j^k = \varepsilon_i^{k-1}, \tag{A62}$$

and in cell j + 1 to the right of the discontinuity boundary, the following values are assigned:

$$p_j^k = p_{i+1}^{k-1}, \ u_j^k = u_{i+1}^{k-1}, \ \rho_j^k = \rho_{i+1}^{k-1}, \ \varepsilon_j^k = \varepsilon_{i+1}^{k-1}, \tag{A63}$$

(2.2 b) if

$$\left| p_{i+1}^{k-1} - p_i^{k-1} \right| \geq e_1$$

then the solution is defined as follows:

-   if

$$p_i^{k-1} > p_{i+1}^{k-1} \text{ and } (x_1 - h/2) \leq x_j \leq (x_1 + h/2),$$

then

$$p_j^k = p_i^{k-1}, \ u_j^k = u_i^{k-1}, \ \rho_j^k = \rho_i^{k-1}, \ \varepsilon_j^k = \varepsilon_i^{k-1}; \tag{A64}$$

-   if

$$p_i^{k-1} < p_{i+1}^{k-1} \text{ and } (x_2 - h/2) \leq x_j \leq (x_2 + h/2),$$

then

$$p_j^k = p_{i+1}^{k-1}, \ u_j^k = u_{i+1}^{k-1}, \ \rho_j^k = \rho_{i+1}^{k-1}, \ \varepsilon_j^k = \varepsilon_{i+1}^{k-1}. \tag{A65}$$

(3)   If

$$(u_{i+1} - u_i) \geq e_1, \tag{A66}$$

then we have the rarefaction wave, and the solution for $x_1 \leq x_j \leq x_2$ is determined from the conservation condition of the Riemann invariants as follows:

$$
\begin{aligned}
u_j^k &= u_i^{k-1} + \frac{u_{i+1}^{k-1} - u_i^{k-1}}{x_2 - x_1}(x_j - x_1), \\
\varepsilon_j^k &= \left( \sqrt{\varepsilon_i^{k-1}} + \frac{\sqrt{\varepsilon_{i+1}^{k-1}} - \sqrt{\varepsilon_i^{k-1}}}{x_2 - x_1}(x_j - x_1) \right)^2, \\
\rho_j^k &= \left( \rho_i^{k-1}\left( \frac{\varepsilon_j^k}{\varepsilon_i^{k-1}} \right)^{\frac{1}{\kappa-1}} + \rho_{i+1}^{k-1}\left( \frac{\varepsilon_j^k}{\varepsilon_{i+1}^{k-1}} \right)^{\frac{1}{\kappa-1}} \right)/2, \\
p_j^k &= (\kappa - 1)\rho_j^k \, \varepsilon_j^k.
\end{aligned}
\tag{A67}
$$

When conditions (1)–(3) are satisfied, if it turns out that the solution has already been assigned to the grid cell at this substep, then it is replaced by a new one when the pressure of the new solution is greater than the pressure of the solution already in the grid cell.

*Appendix B.3. Boundary Conditions for a Wall*

The main idea behind choosing the following boundary conditions is to study the mechanism of wave reflection from the wall. Suppose that, at the acoustics stage, when conditions (1)–(3) are satisfied, the wave $\overline{w}_3 = \{P_R, U_R, R_R, E_R\}$ propagates to the right with velocity $C_R$ and meets the wall, and then it is reflected from it, and transforms into a wave $\overline{w}_1 = \{P_R, -U_R, R_R, E_R\}$ that propagates to the left of the wall with velocity $-C_R$.

In this case, when the wave $\overline{w}_1$ is distributed into the grid cells, the dam-break problem is solved using the Godunov method, where the variables on the left are the data already assigned to the cell, and the variables on the right are $\overline{w}_1 = \{P_R, -U_R, R_R, E_R\}$ As a result, new values of large variables $\{P^*, U^*, R_L^*, E_L^*, D_L^*, R_R^*, E_R^*, D_R^*\}$ are obtained, which are assigned to the grid cells as follows

$$p_i^k = P_i^*, \ u_i^k = U_i^*, \ \rho_i^k = R_{Li}^*, \ \varepsilon_i^k = E_{Ri}^*, \ c_i^k = D_{Li}^* - U_i^*. \tag{A68}$$

For wave $\overline{w}_1 = \{P_L, U_L, R_L, E_L\}$ that, after meeting a wall, turns into the wave $\overline{w}_3 = \{P_L, -U_L, R_L, E_L\}$

$$p_i^k = P_i^*, \ u_i^k = U_i^*, \ \rho_i^k = R_{Ri}^*, \ \varepsilon_i^k = E_{Ri}^*, \ c_i^k = D_{Ri}^* - U_i^*. \tag{A69}$$

At the convection stage, we have one wave that reverses direction when it meets the wall. Under conditions (1)–(3) of the convection stage, wave $\overline{w} = \{p^k, u^k, \rho^k, \varepsilon^k\}$ moves to the right with velocity $u^k$, meets the wall, and is reflected from it, turning into wave $\overline{w} = \{p^k, -u^k, \rho^k, \varepsilon^k\}$, which propagates to the left of the wall with velocity $-u^k$. In this case, when the wave $\overline{w}$ is distributed into the grid cells, the dam-break problem is solved using the Godunov method, where the variables on the left are the data already assigned to the cell, and the variables on the right are $\overline{w} = \{p^k, -u^k, \rho^k, \varepsilon^k\}$. As a result, new values of large variables $\{P^*, U^*, R_L^*, E_L^*, D_L^*, R_R^*, E_R^*, D_R^*\}$ are obtained, which are assigned to the grid cells as shown in (A68). If wave $\overline{w} = \{p^k, u^k, \rho^k, \varepsilon^k\}$ propagates to the left, then a similar reasoning leads to the result described using formula (A69).

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
