# Peer review of "A Semi-Lagrangian Godunov-Type Method without Numerical Viscosity for Shocks"

_fluids, doi:10.3390/fluids7010016_

Round 1

Reviewer 1 Report

The authors provide a study on a semi-Lagrangian Godunov-type Method without Numerical Viscosity for Shocks.  The manuscript is well written, and carries information to the readers. I have, however, some fundamental remarks that if satisfactorily addressed could improve the final impact of the work.

  • The article does not completely contain a Nomenclature table. Please include all symbols in a table dedicated to them, which will be at the beginning of the article after the abstract and keywords. The description of each symbol (e.g., after each equation) may be avoided if a Nomenclature is provided, otherwise, all symbols should be clearly defined at the first instance of appearance in the manuscript.
  • As for the article abstract, it does not describe clearly the considered problem. The abstract should contain answers to the following questions: What problem was studied and why is it important? What methods were used? What are the important results? What conclusions can be drawn from the results? What is the novelty of the work and where does it go beyond previous efforts in the literature? Please include specific and quantitative results in your abstract, while ensuring that it is suitable for a broad audience.
  • The novelty of this study is not clear, and needs to be highlighted explicitly to ensure it is clear to the reader what is new in the research and advancing the state of the art. Simultaneously, it needs to be made clear that the research does not cover only previously known information, and the knowledge gap needs to be clearly addressed.
  • The literature review should be improved to meet the standard of the journal. It does not appear adequate and does not include sufficient important journals.
  • Please provide appropriate references for the mentioned equations. Define all the parameters used in equations.
  • Please determine your future studies in some sentences in conclusion or discussion, to show readers how you want to proceed this work in the future.

Reviewer 2 Report

A certain originality of the present, rather extensive work is certainly recognisable, but also weaknesses in its presentation. Especially section 2, where the finite difference schemes for the individual cases are described in detail, is overlong, which is very tiring even for the most inclined reader. Here I would recommend explaining only the essential ideas and presenting details in an appendix.
Also, I believe that the typical reader of this journal, who is more interested in fluid dynamics in general, is not immediately aware of the advantage of not using numerical viscosity, which is why an explanation of what disadvantages such an approach entails would be desirable.

In contrast to the very long section 2, section 4 is kept very short, although I consider it very important.
Finally, it should be mentioned that besides the Eulerian and Langrangian approaches to the description of shock waves, other approaches exist, for example, the description of such phenomena by means of a variational principle, see Mellmann & Scholle, Symmetry 13(9), 2021, https://www.mdpi.com/2073-8994/13/9/1662, which should also be mentioned.

Round 2

Reviewer 1 Report

The authors addressed the issues I previously pointed out. The revised version of the manuscript is good, and it looks ready for publication.

Reviewer 2 Report

The issues raised by me last time are all addressed, so from my viewpoint the quality of the presentation is at a level at which the manuscript can be published. In particular, the by the additional explanations the originality of the work is highlighted much better.